# Efficient Multi-bit Quantization Network Training via Weight Bias Correction and Bit-wise Coreset Sampling

**Jinhee Kim**[1,2*]   **Jae Jun An**[1*]   **Kang Eun Jeon**[1,3†]   **Jong Hwan Ko**[1†]

[1] Department of Electrical and Computer Engineering, Sungkyunkwan University
[2] Department of Electrical and Computer Engineering, Duke University
[3] Kim Jaechul Graduate School of AI, Korea Advanced Institute of Science and Technology (KAIST)

{a2jinhee,ajj8061,kejeon,jhko}@skku.edu

## Abstract

Multi-bit quantization networks enable flexible deployment of deep neural networks by supporting multiple precision levels within a single model. However, existing approaches suffer from significant training overhead as full-dataset updates are repeated for each supported bit-width, resulting in a cost that scales linearly with the number of precisions. Additionally, extra fine-tuning stages are often required to support additional or intermediate precision options, further compounding the overall training burden. To address this issue, we propose two techniques that greatly reduce the training overhead without compromising model utility: (i) *Weight bias correction* enables shared batch normalization and eliminates the need for fine-tuning by neutralizing quantization-induced bias across bit-widths and aligning activation distributions; and (ii) *Bit-wise coreset sampling strategy* allows each child model to train on a compact, informative subset selected via gradient-based importance scores by exploiting the implicit knowledge transfer phenomenon. Experiments on CIFAR-10/100, TinyImageNet, and ImageNet-1K with both ResNet and ViT architectures demonstrate that our method achieves competitive or superior accuracy while reducing training time up to 7.88×. Our code is released at this link.

## 1   Introduction

With the explosion of highly capable yet computationally demanding deep learning models, quantization has emerged as an effective strategy for balancing performance and efficiency [1, 2, 3, 4]. Despite its advantages, most existing quantization methods are optimized for a single fixed quantization precision configuration, which limits their ability to adapt dynamically to changing resource availability and deployment across various platforms of diverse memory, compute, and power specifications. This has led to a line of recent work focused on training a single model capable of supporting multiple precisions [5, 6, 7, 8], thereby enabling instant adaptation to varying resource budgets at runtime without the need for further training. In such multi-bit quantization networks, henceforth *multi-bit networks*, a single full-precision parent model generates multiple reduced-precision child models, thereby neutralizing the overhead of maintaining separate models for inference. By supporting multiple quantization precisions- referred to as the model's *switchable bit range*, these networks enable adaptive deployment across a wide range of compute-constrained devices [9, 10, 11, 12].

---

[*]Equal contribution.
[†]Corresponding authors.

39th Conference on Neural Information Processing Systems (NeurIPS 2025).

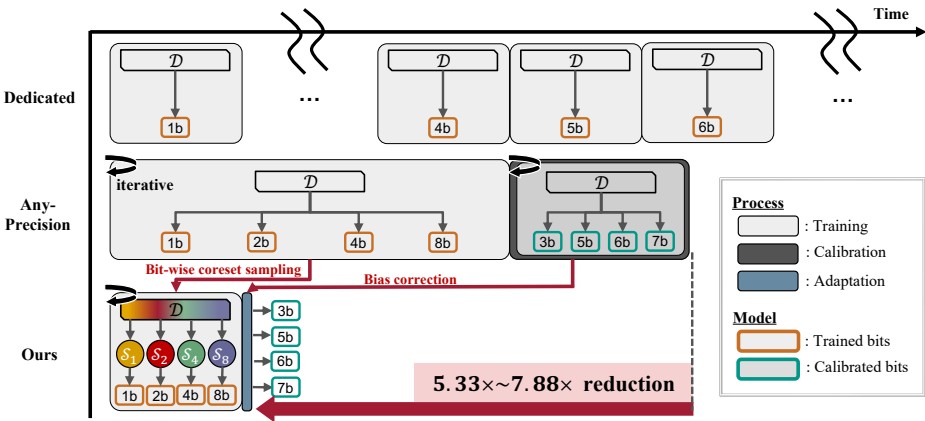

Figure 1: A conceptual diagram of i) Dedicated, ii) Any-Precision, and iii) our training pipelines. $\mathcal{D}$ and $\mathcal{S}$ indicate the full training dataset and coreset, respectively.

Although the multi-bit networks provide flexibility during inference/deployment, this advantage comes at the cost of substantial training overhead, limiting their practical adoption. The prevalent multi-bit training approach, known as *Any-Precision* [13], jointly optimizes the model across a small subset of selected bit-widths, termed the *trained bit range*. While this approach is more efficient than training individual models across the switchable range (*Dedicated*), it still introduces considerable overhead due to the additional calibration required to enable inference at bit-widths outside the trained range (*calibrated bit range*), as shown in Fig. 1. Specifically, calibration demands extensive computation using large amounts of training data to preserve the accuracy of untrained bit models in the calibrated range, by aligning their mismatched activation distributions. We identify that these activation mismatches across different bit-width models stem from biases in the weight distribution induced by quantization. Based on this observation, we propose a novel bias correction technique that directly controls the shift and scaling biases in the quantized weights to align distributions across the entire switchable bit-widths. This alignment enables multiple bit-width sub-networks to share a common set of batch normalization (BN) parameters, effectively eliminating the need for costly post-training calibration.

Another major source of the significant training overhead is the use of the entire dataset for updating models in the trained bit range. Although coreset selection methods have been introduced to reduce the training overhead by identifying a subset of important data samples [14, 15, 16, 17], these approaches have primarily targeted single-precision model training with fixed coresets. Extending this idea to a multi-bit quantization setting, we observe that each bit-width child model can benefit from training on distinct and smaller data subsets due to implicit gradient alignment across bit-widths. Leveraging this insight, we propose a bit-wise coreset sampling method that dynamically selects informative samples individually for each child model, based on the gradients computed per bit-width. Furthermore, since sample importance changes throughout training, we periodically re-sample these coresets to reflect evolving model dynamics. The proposed sampling approach effectively reduces per-epoch computational costs while preserving strong performance through implicit cross-bit-width knowledge transfer, a phenomenon we discover for the first time.

To summarize, our contributions are as follows:

- **Weight bias correction for activation alignment**: We correct quantization-induced biases in the weight space instead of the activation space, enabling multiple child models to share normalization parameters, and in turn eliminating the need for an extra training stage.

- **Bit-wise coreset sampling**: We propose a novel per-bit-width coreset sampling strategy that computes bit-wise importance scores using gradient-based methods, thereby reducing training redundancy in multi-bit quantization networks.

- **Extensive empirical validation**: We demonstrate that our method consistently improves or maintains accuracy while significantly reducing training cost across diverse datasets (e.g., CIFAR-10, CIFAR-100, TinyImageNet, and ImageNet-1K) and architectures (e.g., ResNet, DeiT, Swin). Our method achieves up to 7.88× GPU hour reduction without sacrificing model utility, validating the scalability and generality of our approach.

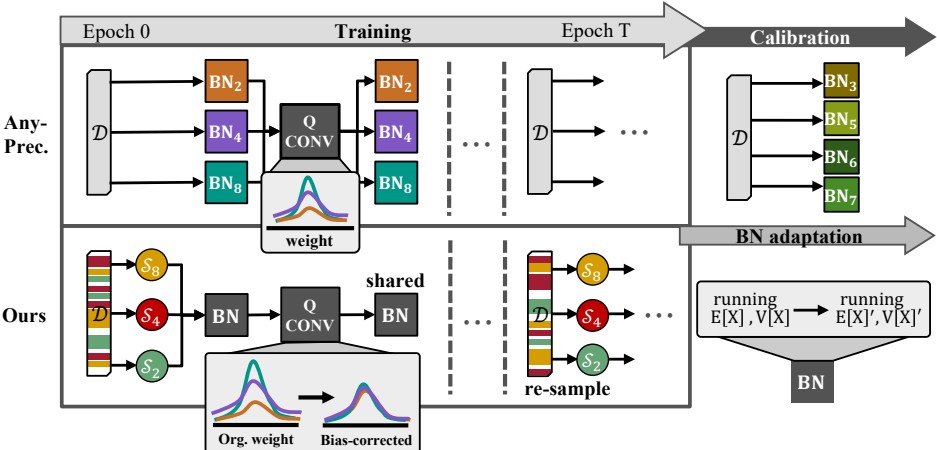

Figure 2: A summary of Any-Precision's and our training pipeline.

## 2 Backgrounds & Related Works

**Multi-bit quantization networks.** Unlike traditional quantized networks that are optimized for a single reduced numerical precision, multi-bit quantization networks [13, 5, 7, 18, 8, 19] are capable of supporting multiple quantization precisions, enabling adaptive and versatile inference/deployment across a wide spectrum of compute-constrained devices [9, 10, 11, 12]. The mainstream approach to training these networks involves optimizing the model for multiple precisions simultaneously, typically by the sum of loss functions corresponding to each bit-width. Formally, this is stated as shown below:

$$\min_{\theta} \sum_{(\mathbf{x},y)\in\mathcal{S}} \sum_{b\in\mathcal{B}} \mathcal{L}(\mathbf{x}, y, Q(\theta, b)), \tag{1}$$

where $\theta \in \mathbb{R}^d$ denotes the learnable model parameter which is shared across multiple precisions; $\mathcal{L}(\mathbf{x}, y, Q(\theta, b))$ is the loss on training sample $(\mathbf{x}, y)$ in training set $\mathcal{S}$; $Q(\theta, b) \in \mathbb{Z}^d$ is the quantized version of $\theta$ at $b$-bit precision; and $\mathcal{B}$, referred to as the ***trained range***, represents the set of all ***trained bit-widths***. To perform this optimization in practice, the batch-wise training scheme, which interleaves parameter updates across child models corresponding to different bit-widths in a batch-wise manner (see Algorithm 1), is commonly adopted to promote generalization across the entire training range.

**Training overhead of multi-bit networks.** While multi-bit training is generally more efficient than training multiple single-precision networks independently, it still incurs significant computational overhead—particularly as the training range $\mathcal{B}$ expands. To mitigate this cost, recent approaches minimize the number of bit-widths included in the training range and instead introduce ***calibrated bit-widths/range*** to expand precision support. Specifically, the model is first trained on a small set of bit-widths (the trained range), after which a large portion of its parameters are frozen. The remaining parameters are then calibrated or lightly fine-tuned to support additional bit-widths (the calibrated range). The union of the trained and calibrated ranges defines the model's ***switchable range***, $\mathcal{R}$—i.e., the full set of bit-widths supported by the multi-bit network.

---

**Algorithm 1** Batch-wise training scheme

---

**Input:** Data $\mathbf{X}$, label $\mathbf{Y}$
**Output:** Multi-bit network G

1: **for** epoch = 1, ..., T **do**
2:     **for** batch from $\mathbf{X}$, $\mathbf{Y}$ **do**
3:         **for** bit $b$ in $\mathcal{B}$ **do**
4:             Set all layers in G to $b$-bit
5:             Compute forward pass of G
6:             Calculate gradients of G
7:         **end for**
8:         Update parameters with $\sum_{\mathcal{B}} \mathcal{L}_b$
9:     **end for**
10: **end for**

---

**Challenges in training multi-bit networks.** One major challenge in multi-bit network training research is the accuracy degradation due to activation distribution mismatch between different bit-widths. To address this mismatch, Any-Precision [13], along with CoQuant [18] and MBQuant [8], leverages the '*switchable batch normalization*' approach first proposed by [20]. While effective, assigning separate batch normalization layers to each bit-width incurs additional overhead: obtaining

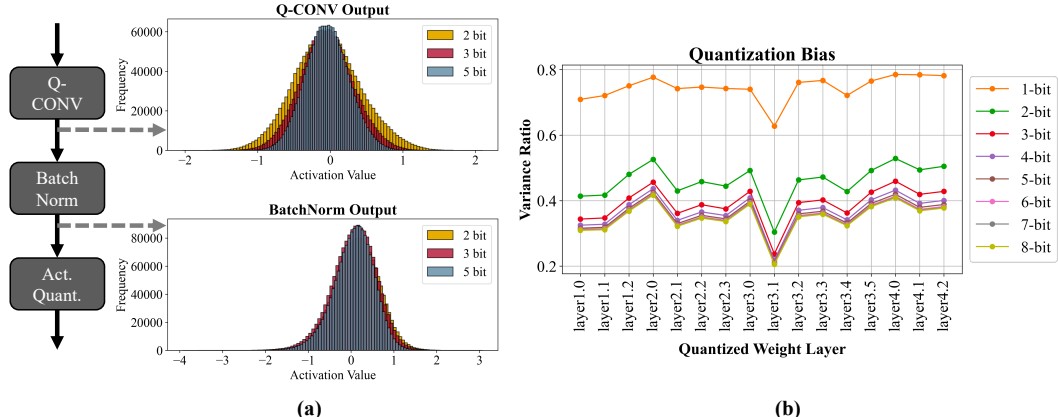

Figure 3: (a) Mismatch in activation distributions between different bit-widths, and (b) variance ratio between quantized weights and original weights in ResNet-50.

these parameters for unseen bit-widths typically requires an extra training phase as shown in Fig. 2. Some recent works [7, 5, 21] avoid this mechanism altogether, but often suffer from degraded performance at lower bit-widths or resort to computationally expensive strategies to reduce interference between the training objectives of different child models. In our proposed method, rather than relying on multiple batch normalization layers or other costly techniques, we correct the weight distribution directly before and after quantization. Our key insight is that quantization introduces bit-width-specific shifts and scaling in the weight distribution, and aligning these distributions helps reduce conflicts across child models during training.

**Coreset selection.** Coreset selection—also known as dataset pruning—aims to reduce model training cost by selecting a small yet representative subset of the training data, while preserving model utility. The core challenge lies in accurately identifying the most informative samples. Feature-space based methods select subsets that preserve the geometry of the data distribution-for example, Herding [22] and Moderate [15] select data points with distance in feature space. Uncertainty-based methods prioritize ambiguous or hard-to-classify samples; for example, Entropy [23] and Cal [24] select samples near decision boundaries. Gradient-based methods leverage training loss gradients. GraND/EL2N [14] rank samples by their gradient magnitude (or prediction error), while Craig [25] and GradMatch [26] select subsets that best match/mimic the full dataset's gradient signals. Training-dynamics based methods consider samples' behavior over many epochs. Forgetting [16] counts how often a sample is forgotten during training, and AUM [27] averages the confidence gap across all epochs. Finally, hybrid approaches fuse multiple criteria: TDDS [17] integrates gradient information with training dynamics by measuring each sample's variability in its epoch-wise contribution to the overall training gradient.

**Shortcomings of existing coreset selection research.** Despite this breadth of approaches, most coreset selection methods are investigated under the assumption of fixed, full-precision floating-point models, and their applicability to quantized neural networks remains largely unexplored. In particular, the integration of coreset selection into quantization-aware training (QAT) has received little attention, let alone its extension to the more complex setting of *multi-bit quantization*, where cross-bit interactions can significantly affect saliency estimation and the underlying training dynamics.

## 3 Weight Bias Correction for Activation Alignment

**Activation distribution mismatch in multi-bit networks.** As discussed in Section 2, multi-bit networks often suffer from mismatched activation distributions across bit-widths. To isolate the source of this mismatch, we decompose the post-convolutional activation into two components: the quantized input activation and the quantized weight. To simplify the analysis, we fix the input activation to a specific precision (e.g., 4-bit). Under this setup, any observed variation in the output can be attributed solely to the quantized weights, thereby reducing the problem to a single source of quantization noise.

In Fig. 3(a), we visualize the post-convolutional activations of ResNet-50 for bit-widths $b \in \mathcal{B} = \{2, 3, 5\}$ using a batch of ImageNet examples. Although the input activation is fixed, the output distributions vary noticeably across bit-widths. This indicates that the differences can be attributed to quantization-induced bias in the weights. Fig.3(b) supports this explanation by showing that the quantized weights exhibit clear scale distortions compared to the original weights. This observation is consistent with prior observations of [2, 28], which highlight the presence of systematic bias introduced during quantization.

Many multi-bit networks [13, 18, 8] address this activation mismatch problem by training separate BN parameters for each bit-width to independently correct activation distributions. As illustrated in Fig. 3(a), this approach successfully aligns BN outputs across different precisions. While effective, aligning output activations typically requires access to the training data and additional forward/backward passes, which incurs additional training overhead.

**Bias correction for quantized weights.** Instead of rectifying the output activations, we address the bias at its source by aligning the quantized weights prior to convolution. By doing this, we can match activation outputs across bit-widths just by correcting the weights during the initial training stage, without having to explicitly match the activations themselves. As a result, BN layers can be shared across all bit-widths, as shown in Fig. 2, avoiding the additional overhead of calibrating separate BN layers. It is important to note that this correction is performed under a fixed activation bit-width (e.g., 4 bits), meaning that aligning the weights directly translates to more consistent activation outputs across different bit-widths. Specifically, we adjust the quantized weight vector $\mathbf{w}_q$ with respect to their full-precision counterpart $\mathbf{w}$, and compute the corrected weights $\mathbf{w}_q'$ as follows:

$$\mathbf{w}_q' = \sqrt{\frac{\mathbb{V}[\mathbf{w}]}{\mathbb{V}[\mathbf{w}_q]}}(\mathbf{w}_q + (\mathbb{E}[\mathbf{w}] - \mathbb{E}[\mathbf{w}_q])), \tag{2}$$

where $\mathbb{E}[\cdot]$ and $\mathbb{V}[\cdot]$ denote the expectation and variance, respectively. This weight alignment enables multiple child models to share a single set of BN parameters with minimal interference. To compensate for residual discrepancies not fully addressed by bias correction, we additionally apply BN adaptation [29]. While adjusting running statistics has been proven to be effective in fixed-quantized networks [30], its use in multi-bit networks [5] remains limited and often lacks clarity on when it is applied (e.g., applied at every epoch in [5], which is unnecessary). Applying BN adaptation once at the final training stage as shown in Fig. 2, is sufficient to correct the running mean and variance for each bit-width, achieving optimal performance without additional overhead.

## 4 Bit-Wise Coreset Sampling

To translate the benefits of coreset selection into multi-bit quantization networks, we propose two techniques tailored to this setting: (i) a coreset sampling strategy that accounts for variations in sample importance across bit-widths and training epochs; and (ii) a bit-wise training scheme for accurate per-bit-width importance score evaluation. Together, these techniques enable more efficient and adaptive training across a range of quantization levels while maintaining strong model utility.

### 4.1 Coreset sampling strategy

The central idea behind our coreset sampling method is to dynamically redraw training subsets along two axes: bit-width and training time. Rather than using a static, global coreset, we select bit-wise coresets that evolve throughout training via sampling as shown in Fig. 2. This design is motivated by two key observations: (i) gradient alignment across bit-widths, and (ii) temporal drift in sample importance.

**Observation 1 – Gradient alignment across bit-widths.** We find that gradients computed from different bit-widths using the same data sample are highly aligned. In Fig. 4, we visualize the angles between the gradients of the 8-bit and 2-bit child models across several layers of ResNet-20 at various training epochs. It can be seen that the angle between the two gradients stays consistently below $28°$, with alignment improving in deeper layers. This implies that, without loss of generality, parameter updates based on 2-bit gradients positively influence 8-bit child model (and also that of other precisions in the trained range), and vice versa. We refer to this phenomenon as *cross-bit-width implicit knowledge transfer*, where shared parameters act as conduits for the transfer of learning signals between child models.

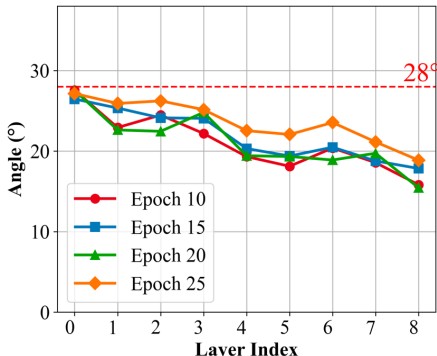

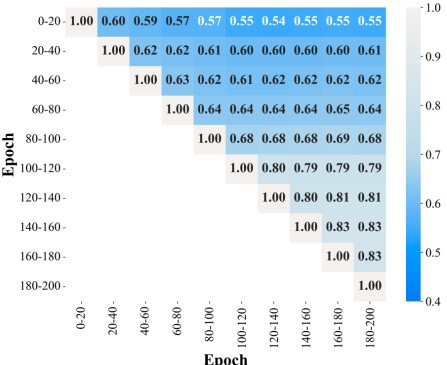

Figure 4: Angle between the 8-bit and 2-bit gradients across layers.

Figure 5: Spearman correlation between ranks at different epochs.

This observation leads to a key insight: *it is unnecessary to feed the same sample to all bit-widths during training*. Accordingly, we construct bit-wise coresets—separate subsets tailored for each bit-width. These bit-wise coresets not only reflect the variation in sample importance across bit-widths but also exploit the implicit gradient transfer phenomena to benefit from a collective learning signal without accessing the full dataset. Therefore, this design significantly reduces per-epoch computational cost while preserving strong performance across the trained range.

**Observation 2 – Temporal drift in sample importance.** We also observe that a sample's importance evolves as training progresses. More specifically, samples influential in early epochs often become less relevant in later stages, where the model nears convergence and the loss landscape flattens. In Figure 5, we visualize this effect using Spearman correlation of TDDS-based importance scores measured at different training stages on ResNet-18 trained with CIFAR-100 dataset. Correlations between early and late-stage scores may drop as low as 0.54, indicating substantial shifts in sample influence over time.

To account for the temporal drift in sample importance, we periodically re-sample each bit-wise coreset throughout training. Although high-score samples may be informative in the early stages, they often lose relevance as the model's learning dynamics evolve. Continually refreshing the coresets helps prevent overfitting to outdated importance estimates, especially critical when the true sample importance landscape is dynamic and only partially observable.

**Sampling method.** To construct the bit-wise coresets via sampling, we first convert the sample importance scores into *sampling probabilities* by applying min-max normalization. We then further shape the sampling probabilities using temperature-based sampling [31, 32, 33], which simultaneously reduces overfitting to high-scoring samples and promotes diversity, effectively balancing exploitation and exploration. The sampling probability $p_i^{(b)}$ for sample $i$ at bit-width $b$ is defined as:

$$p_i^{(b)}(\tau) = \frac{\left(s_i^{(b)}\right)^{1/\tau}}{\sum_{\forall j}\left(s_j^{(b)}\right)^{1/\tau}}, \tag{3}$$

where $s_i^{(b)}$ denotes the min-max normalized importance score for sample $i$ at bit-width $b$; and $\tau > 0$ denotes the temperature parameter. Note that importance scores are computed once prior to coreset sampling and remain fixed throughout training.

## 4.2 Bit-wise training scheme for score evaluation

Extracting accurate, bit-wise importance scores is particularly challenging in the context of training dynamics-based coreset selection methods. These mainstream approaches estimate sample importance over multiple training epochs to capture the intricate training dynamics and improve score reliability (see Section 2 for an overview). A representative example is TDDS [17], which accumulates intermediate gradient signals—referred to as context vectors—throughout training to capture the evolving contribution of each sample. While effective in single-precision settings, we find that applying such methods directly under the standard batch-wise training scheme (Algorithm 1) fails to produce meaningful bit-wise importance estimates. The core issue lies in the interleaved update

Table 1: ResNet on CIFAR-10 and 100. Pruning rate of *Coreset Sampling* is 80%.

| Dataset | Framework | Coreset Sampling | Test Accuracy | | | | | | GPU hours (Speed up) |
| | | | 1bit | 2bit | 4bit | 8bit | 32bit | Avg. | |
|---|---|---|---|---|---|---|---|---|---|
| CIFAR-10 | Dedicated | - | 92.42 | 93.04 | 92.99 | 93.08 | 94.11 | 93.10 | 11.97 (1.00×) |
| | Any-Prec. | - | 92.85 ±0.21 | 93.28 ±0.17 | 93.61 ±0.07 | 93.64 ±0.04 | 93.77 ±0.12 | 93.31 | 8.76 (1.36×) |
| | Ours | - | 93.11 ±0.07 | 93.46 ±0.11 | 93.57 ±0.08 | 93.53 ±0.05 | 93.60 ±0.05 | **93.46** | 7.52 (**1.59×**) |
| | | ✔ | 92.60 ±0.14 | 93.01 ±0.10 | 93.03 ±0.04 | 93.00 ±0.13 | 93.08 ±0.12 | 92.97 | 1.52 (**7.88×**) |
| CIFAR-100 | Dedicated | - | 67.52 | 70.21 | 70.17 | 70.50 | 72.63 | 70.21 | 11.19 (1.00×) |
| | Any-Prec. | - | 70.54 ±0.31 | 71.54 ±0.27 | 71.60 ±0.27 | 71.58 ±0.39 | 72.23 ±0.29 | 71.47 | 8.27 (1.35×) |
| | Ours | - | 70.95 ±0.09 | 71.92 ±0.17 | 71.96 ±0.11 | 71.91 ±0.09 | 71.91 ±0.08 | **71.84** | 7.17 (**1.56×**) |
| | | ✔ | 69.14 ±0.08 | 70.12 ±0.11 | 70.35 ±0.17 | 70.43 ±0.11 | 70.41 ±0.11 | **70.26** | 1.47 (**7.61×**) |

Table 2: ResNet on CIFAR-10. Comparison against previous methods at 80% pruning rate.

| Method | Test Accuracy | | | | | |
| | 1bit | 2bit | 4bit | 8bit | 32bit | Avg. |
|---|---|---|---|---|---|---|
| Random | 88.97 ±0.47 | 89.99 ±0.22 | 90.21 ±0.40 | 90.12 ±0.28 | 89.68 ±0.32 | 89.94 |
| Entropy | 85.62 ±0.04 | 86.20 ±0.39 | 86.31 ±0.24 | 86.25 ±0.22 | 86.36 ±0.15 | 86.21 |
| Forgetting | 76.57 ±1.10 | 78.18 ±0.97 | 78.46 ±0.99 | 78.36 ±1.00 | 78.49 ±0.90 | 78.14 |
| EL2N | 80.21 ±0.29 | 81.16 ±0.14 | 81.21 ±0.20 | 81.15 ±0.43 | 81.22 ±0.12 | 81.07 |
| Moderate | 87.63 ±0.22 | 88.18 ±0.18 | 88.35 ±0.11 | 88.43 ±0.21 | 88.27 ±0.08 | 88.26 |
| TDDS | 87.67 ±0.55 | 88.35 ±0.05 | 88.72 ±0.09 | 88.75 ±0.07 | 88.57 ±0.19 | 88.54 |
| **Ours** | 92.60 ±0.14 | 93.01 ±0.10 | 93.03 ±0.04 | 93.00 ±0.13 | 93.08 ±0.12 | **92.97** |

pattern: gradients from all bit-widths are aggregated before a shared parameter update, resulting in a single unified context vector that masks the distinct training dynamics of each child model.

To address this issue, we introduce a bit-wise training scheme for score evaluation, as shown in Algorithm 2. In this setup, each child model corresponding to a trained bit-width is trained on the entire dataset before proceeding to the next bit-width. This scheduling isolates the gradient updates for each precision, enabling the computation of distinct context vectors and more accurate, bit-wise importance scores.

It is important to note that the proposed bit-wise training scheme is used *exclusively* for importance score extraction. Once the scores are computed, we revert to the standard batch-wise training scheme for actual multi-bit network training. This hybrid approach allows coreset construction to benefit from accurate, bit-wise decoupled importance evaluation while preserving the generalization advantages of batch-wise training.

---

**Algorithm 2** Bit-wise training scheme

**Input:** Data $\mathbf{X}$, label $\mathbf{Y}$
**Output:** Multi-bit network G

1: **for** epoch = 1, ..., T **do**
2:     **for** bit $b$ in $\mathcal{B}$ **do**
3:         **for** batch from $\mathbf{X}$, $\mathbf{Y}$ **do**
4:             Set all layers in G to $b$-bit
5:             Compute forward pass of G
6:             Calculate gradients of G
7:             Update parameters with $\mathcal{L}_b$
8:         **end for**
9:     **end for**
10: **end for**

---

## 5 Experiments

### 5.1 Setup

**Evaluation metrics and baselines.** We evaluate our method in terms of per-bit-width accuracy and total GPU hours. Comparisons are made against: (1) the dedicated framework, (2) the standard multi-bit framework (e.g., Any-Precision [13]), and (3) our method which augments the standard framework with Bias Correction (***Ours***). Within ***Ours***, we evaluate our proposed *coreset sampling* strategy, which uses *bit-wise scores*, against six baseline corset selection methods: Random, Entropy [23], Forgetting [16], EL2N [14], Moderate [15], and TDDS [17]. Since most existing coreset selection techniques are designed for dedicated training, we adapt each baseline to our multi-bit framework to ensure a fair comparison.

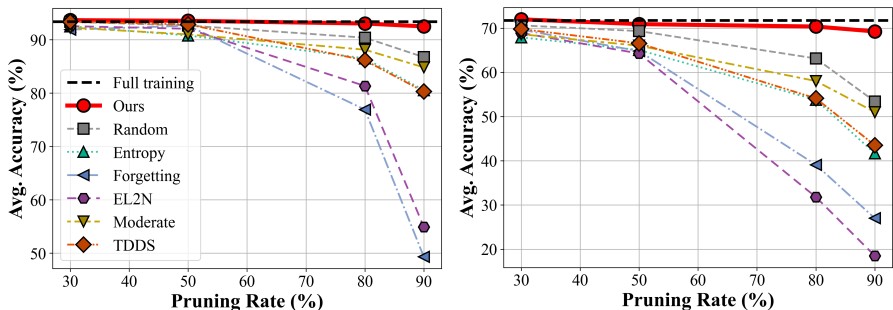

Figure 6: Accuracy comparison across different pruning rates. *Left*: CIFAR-10, *Right*: CIFAR-100

Table 3: ResNet on ImageNet-1K. Pruning rate is 80%.

| Framework | Coreset Sampling | Test Accuracy | | | | | | GPU hours (Speed up) |
| | | 1bit | 2bit | 4bit | 8bit | 32bit | Avg. | |
|---|---|---|---|---|---|---|---|---|
| Dedicated | - | 57.93 | 68.74 | 74.12 | 74.96 | 75.95 | 72.04 | 39.91 (1.00×) |
| Any-Prec. | - | 68.77 | 71.66 | 73.84 | 74.07 | 74.63 | 73.01 | 33.94 (1.18×) |
| Ours | - | 68.12 | 72.34 | 73.97 | 74.20 | 74.32 | **73.22** | 27.21 **(1.47×)** |
| | ✔ | 67.22 | 71.93 | 73.29 | 73.40 | 73.92 | **72.51** | 6.94 **(5.75×)** |

Table 4: Accuracy comparison against previous methods at 80% pruning.

| Method | Test Accuracy | | | | | |
| | 1bit | 2bit | 4bit | 8bit | 32bit | Avg. |
|---|---|---|---|---|---|---|
| Random | 66.13 | 70.68 | 72.07 | 72.05 | 72.97 | 71.36 |
| Entropy | 65.32 | 69.28 | 70.66 | 70.77 | 71.68 | 70.03 |
| Forgetting | 55.79 | 60.70 | 64.87 | 65.48 | 66.78 | 63.73 |
| EL2N | 63.19 | 68.17 | 69.64 | 69.75 | 70.72 | 68.90 |
| Moderate | 64.57 | 68.88 | 70.39 | 70.63 | 71.58 | 69.74 |
| TDDS | 65.24 | 69.21 | 70.56 | 70.71 | 71.36 | 69.42 |
| **Ours** | 67.22 | 71.93 | 73.29 | 73.40 | 73.92 | **72.51** |

**Datasets and networks.** We evaluate our method on four canonical datasets—CIFAR-10, CIFAR-100 [34], TinyImageNet [35], and ImageNet-1K [36]—with a diverse set of networks. These include three ResNet models: PreActResNet-20 [37], ResNet-18, and ResNet-50 [38], as well as three Vision Transformers (ViTs): DeiT-T, DeiT-S [39], and Swin-T [40].

**Implementation details.** All experiments are conducted on a single NVIDIA A100 GPU. Each experimental setting is as follows: (i) *Dedicated* trains the model with a single weight and activation bit-width. (ii) *Any-Precision* uses a training range of $\mathcal{B} = 1, 2, 4, 8, 32$ for ResNet models (and $\mathcal{B} = 2, 8$ for ViTs). After training, independent parameters for the remaining bit-widths are calibrated for approximately one-third of the training epochs to ensure convergence. (iii) *Bias Correction* adopts the same training range as (ii) but skips the calibration phase and instead performs BN adaptation. Specifically, we assign a separate BN layer for 1-bit and share BN layers for all other bit-widths. Although inference for the calibrated range is supported and achieves accuracy comparable to the trained range in both (ii) and (iii), those results are omitted here due to the page limit and are provided in the Appendix.

## 5.2 Results

**ResNet on CIFAR-10 and 100.** Table 1 presents the results on PreActResNet-20 for CIFAR-10 and ResNet-18 for CIFAR-100, highlighting both performance and training time reduction achieved by our method. Compared to existing baselines, *Ours* achieves competitive performance with reduced training time by eliminating the calibration phase, while *coreset sampling* further improves efficiency by reducing data usage without compromising accuracy. Additional results for CIFAR-10 with PreActResNet-20, including comparisons between our coreset sampling method and six baselines (see Section 5.1), are presented in Table 2. Random selection has been shown to excel at high pruning rates in prior studies [17], and we observe the same trend in our experimental setup. Our method shows consistent improvements in both accuracy and efficiency across bit-widths. Figure 6 shows the trade-off between training cost and accuracy by plotting average accuracy against pruning rate. Our method consistently outperforms all baselines across the entire pruning spectrum and maintains high accuracy even at a 90% pruning rate.

**ResNet on ImageNet-1K.** We further demonstrate the effectiveness of our method for a bigger dataset like ImageNet-1K. Table 3 and Table 4 summarize our results for ResNet-50 on ImageNet-1K with respect to the baseline and previous methods. For these experiments, we finetune from a pretrained Any-Precision model, where specific implementation details are provided in the Appendix.

Table 5: DeiT-T, DeiT-S, Swin-T on ImageNet-1K for different pruning rates.

| Network | Framework | Pruning Rate | Test Accuracy | | | | | | | | GPU hours (Speed up) |
|---|---|---|---|---|---|---|---|---|---|---|---|
| | | | 2 | 3 | 4 | 5 | 6 | 7 | 8 | Avg. | |
| **DeiT-T** | Any-Prec. | - | - | 69.72 | 69.97 | 70.59 | 70.77 | 70.86 | 70.91 | 70.47 | 25.77 (**1.00×**) |
| | Ours | 50% | - | 69.30 | 69.42 | 69.93 | 70.15 | 70.22 | 70.16 | **69.86** | 10.00 (**2.58×**) |
| | | 60% | - | 69.03 | 68.87 | 69.65 | 69.99 | 69.93 | 70.00 | **69.58** | 7.55 (**3.41×**) |
| **DeiT-S** | Any-Prec. | - | 76.34 | 76.93 | 78.19 | 78.25 | 78.30 | 78.32 | 78.37 | 77.81 | 27.07 (**1.00×**) |
| | Ours | 50% | 76.05 | 76.45 | 78.04 | 78.21 | 78.16 | 78.18 | 78.15 | **77.61** | 13.33 (**2.03×**) |
| | | 60% | 76.22 | 76.59 | 78.05 | 78.25 | 78.36 | 78.27 | 78.31 | **77.72** | 11.67 (**2.32×**) |
| **Swin-T** | Any-Prec. | - | 78.68 | 79.14 | 79.86 | 79.97 | 79.96 | 79.94 | 79.96 | 79.64 | 27.10 (**1.00×**) |
| | Ours | 50% | 78.49 | 78.90 | 79.79 | 79.92 | 79.88 | 79.93 | 79.96 | **79.55** | 13.27 (**2.04×**) |
| | | 60% | 78.53 | 78.92 | 79.67 | 79.88 | 79.85 | 79.94 | 79.90 | **79.53** | 10.57 (**2.56×**) |

Table 6: The effect of Bias Correction and BN Adaptation.

| Dataset | Bias Correction | BN Adaption | Test Accuracy | | | | | | | | | |
|---|---|---|---|---|---|---|---|---|---|---|---|---|
| | | | 1bit | 2bit | 3bit | 4bit | 5bit | 6bit | 7bit | 8bit | 32bit | Avg. |
| **CIFAR-10** | - | - | 92.95 | 87.72 | 93.53 | 93.32 | 92.73 | 92.47 | 92.08 | 91.87 | 93.53 | **92.24** |
| | ✔ | - | 93.58 | 91.98 | 93.50 | 93.68 | 93.62 | 93.51 | 93.47 | 93.40 | 93.70 | **93.38** |
| | - | ✔ | 92.87 | 93.36 | 93.59 | 93.55 | 93.56 | 93.52 | 93.56 | 93.63 | 93.65 | **93.48** |
| | ✔ | ✔ | 93.61 | 93.72 | 93.88 | 93.89 | 93.92 | 93.84 | 93.88 | 93.83 | 93.92 | **93.83** |
| **CIFAR-100** | - | - | 70.23 | 53.18 | 70.48 | 70.88 | 69.52 | 68.19 | 67.25 | 66.88 | 70.21 | **67.42** |
| | ✔ | - | 71.12 | 69.03 | 71.58 | 72.03 | 71.63 | 71.39 | 71.23 | 71.10 | 71.83 | **71.22** |
| | - | ✔ | 70.36 | 70.95 | 71.53 | 71.56 | 71.40 | 71.46 | 71.52 | 71.47 | 71.45 | **71.30** |
| | ✔ | ✔ | 71.37 | 72.10 | 72.31 | 72.37 | 72.27 | 72.34 | 72.38 | 72.33 | 72.26 | **72.19** |

Compared to the dedicated training setting, our method substantially reduces training time by $5.75\times$ with minimal impact on performance.

**ViTs on ImageNet-1K.** We also evaluate our method on larger transformer-based models to demonstrate the generality and scalability of our method to other architectures. The results of three different ViTs (i.e., DeiT-T, DeiT-S, and Swin-T) are summarized in Table 5. To the best of our knowledge, there is not yet a standard multi-bit framework such as Any-Precision for vision transformers. To this end, we implement our own framework with similar configurations as Any-Precision. We compare our method with two dataset pruning ratios- 50% and 60% respectively. With a bigger dataset, our method shows even more significant reduction in training time (as large as 18.22 GPU hours reduction in DeiT-T), while showing consistent accuracy compared to the baselines.

## 5.3 Ablation

**Effect of Bias Correction and BN Adaptation.** As shown in Table 6, we conduct an ablation study to quantify the individual and combined effects of Bias Correction and BN adaptation in the final training stage. The results show that both components play distinct yet complementary roles in achieving stable alignment and high accuracy across bit widths. Bias Correction primarily compensates for systematic deviations introduced during quantization, restoring the representational balance in the weight space. However, since it does not modify the batch normalization statistics used at inference, it alone cannot fully align the activation distributions. BN adaptation, applied at the final stage, addresses this limitation by recalibrating the running mean and variance through a small number of forward passes, thereby aligning post-quantization activations with their floating-point counterparts. Together, these two procedures act on different levels of the model—weight and activation—resulting in consistent improvements across all bit widths. Quantitatively, the combination yields the highest accuracy, confirming that the final BN adaptation provides additional activation alignment beyond what bias correction alone can achieve. These findings clarify the respective contributions of both components and highlight the importance of performing BN adaptation at the last training stage for precise activation calibration in quantized models.

Table 7: The effect of the bit-wise training scheme.

| Pruning | CIFAR-10 | | CIFAR-100 | |
| Rate | Batch-wsie | **Bit-wise** | Batch-wise | **Bit-wise** |
|---|---|---|---|---|
| **70%** | 90.89 | 91.17 (**0.29↑**) | 61.13 | 64.06 (**2.93↑**) |
| **80%** | 88.55 | 88.91 (**0.36↑**) | 54.23 | 59.51 (**5.28↑**) |
| **90%** | 80.27 | 82.79 (**2.52↑**) | 43.52 | 48.87 (**5.35↑**) |

**Effect of bit-wise schedule for score extraction.** To quantify the benefit of isolating per–bit-width dynamics, we perform a fixed-coreset ablation under the multi-bit framework (i.e., Any-Precision [13]) with Bias Correction setting. We select the dataset once—either by (i) the conventional batch-wise 1 TDDS scores [17] or by (ii) our bit-wise 2 scores. We then train the full multi-bit schedule on these reduced sets. As reported in Table 7, bit-wise scoring yields higher accuracy at every pruning ratio on both datasets. This improvement stems from our bit-wise extraction design 2, which enables the collection of separate intermediate gradients per bit-width at each epoch—allowing us to compute distinct context vectors for every sub-model.

## 6 Conclusion

In this work, we introduce two techniques to reduce the training overhead of multi-bit quantization networks. First, we correct quantization-induced bias in the weight space, removing the need for an additional training stage. Second, we design a bit-wise coreset sampling strategy that leverages implicit knowledge transfer, allowing each child model to train on a compact, informative subset selected via gradient-based importance scores. Our approach preserves model utility while reducing training costs across various architectures such as ResNets and ViTs, offering a scalable solution for efficient multi-bit quantization training. By enabling more efficient multi-precision learning, our method contributes to the broader goal of sustainable and energy-efficient AI, helping make high-performance models more accessible, affordable, and ubiquitous to everyone.

Despite strong empirical results, our evaluations are limited to computer vision tasks due to the high computational cost of training multi-bit networks—a challenge shared by every prior work in this space. By significantly reducing this overhead, our method paves the way for applying multi-bit quantization to more demanding applications such as generative AI and large-scale language tasks. Extending our approach to these domains will be the focus of our future work, advancing the broader applicability and impact of multi-bit quantization networks across diverse tasks.

## 7 Acknowledgment

We thank the anonymous reviewers for their constructive comments. This work was partly supported by the National Research Foundation of Korea (NRF) grant (RS-2024-00345732, RS-2025-02216217); the Institute for Information & communications Technology Planning & Evaluation (IITP) grants (RS-2020-II201821, RS2019-II190421, RS-2021-II212052, RS-2021-II212068, RS2025-02217613, RS-2025-10692981, RS-2025-25442569); the Technology Innovation Program (RS-2023-00235718, 23040-15FC) funded by the Ministry of Trade, Industry & Energy (MOTIE, Korea) grant (1415187505).

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

# Appendix

# Contents

# A   Implementation Details

**Quantization function.** For CNN experiments, we quantize the weights and activations following the baseline settings of Any-Precision [13], where the activation bit-width is fixed at 4-bits. Any-Precision employs the DoReFa [41] quantizer into its framework. The quantization process is as follows:

$$s = \mathbb{E}(|\mathbf{W}|) \tag{4}$$

$$\mathbf{W}' = \frac{\tanh(\mathbf{W})}{2\max(|\tanh(\mathbf{W})|)} + 0.5 \tag{5}$$

$$\bar{\mathbf{Q}}_n = \lceil M_n \mathbf{W}' \rfloor \tag{6}$$

$$\mathbf{Q}_n = s(2\bar{\mathbf{Q}}_n/M_n - 1) \tag{7}$$

where $\mathbf{W}'$ is the normalized weight tensor; $\bar{\mathbf{Q}}_n$ is the quantized bin; and $\mathbf{Q}_n$ is the quantized floating-point weight. $M_n = 2^n - 1$ is the maximum bin value with respect to the target bit-width $n$.

For ViT experiments, we adopt the StatsQ quantizer proposed in [42] for weight quantization. Activation quantization is performed using the LSQ quantizer [43], fixed at 8-bits. StatsQ applies uniform quantization, using the channel-wise (or head-wise) mean of the absolute weight values as

the scaling factor. The quantization process is defined as follows:

$$s = 2\mathbb{E}_{\mathcal{A}}(|\mathbf{W}|) \tag{8}$$

$$\mathbf{W}' = \mathrm{clip}(\mathbf{W}/s, -1, 1) \tag{9}$$

$$\mathbf{W}'' = \frac{\mathbf{W}'}{2\max(|\mathbf{W}'|)} + 0.5 \tag{10}$$

$$\bar{\mathbf{Q}}_n = \lceil M_n \mathbf{W}'' \rfloor \tag{11}$$

where the remaining steps for mapping the quantized values back to the floating-point range follow the same procedure as in DoReFa. Here, the expectation in the scaling factor $s$ is taken over dimension(s) $\mathcal{A}$, which depend on the shape of $\mathbf{W}$: for 2D weights, $\mathcal{A}$ corresponds to the column dimension; for 3D weights, it includes both the first and last dimensions.

**Implementation of Any-Precision ViTs.** To the best of our knowledge, there is currently no standardized multi-bit framework like Any-Precision [13] available for vision transformers (ViTs). To address this, we develop our own framework, adopting configurations similar to those used in Any-Precision. Our implementation is based on the QAT framework from the OFQ [42] codebase, with minor modifications to the quantizer to match the quantization function described above and without the query-key reparameterization proposed in [42].

**Configurations of baseline coreset selection methods.** To evaluate the effectiveness of our method, we compare it against several existing coreset selection methods. While prior approaches typically focus on selecting a single subset optimized for a single model or configuration, our method targets bit-wise unique subsets tailored for multi-bit training. Accordingly, we apply each baseline's scoring strategy independently to each sub-model, constructing distinct coresets, and train each sub-model on its respective subset. For CIFAR-10 and CIFAR-100, we train for 200 epochs; for ImageNet-1K, the number of epochs varies depending on the model. For ImageNet-1K experiments, we initialize from pretrained models: ResNet models use the Any-Precision weights [13], while ViTs are initialized from the 8-bit quantized checkpoints provided by PTQ4ViT [44]. The specific configurations for each coreset selection method are as follows:

- **Entropy** [23], **EL2N** [14]: Scores are computed over 20 epochs for CIFAR-10/100 and 5 epochs for ImageNet-1K. Entropy identifies samples near decision boundaries, while EL2N ranks samples based on the magnitude of their gradients.

- **Forgetting** [16]: Scores are computed over 200 epochs for CIFAR-10/100 and 10 epochs for ImageNet-1K. A forgetting score is defined by the number of times a sample is misclassified after being learned correctly, capturing how often a sample is "forgotten" during training.

- **Moderate** [15]: Scores are computed using features extracted by a pre-trained model for 200 epochs on CIFAR-10/100, and for 70 epochs on ImageNet-1K, as an importance proxy. Moderate quantifies importance by measuring distances between samples in the resulting feature space.

- **TDDS** [17]: Scores are computed over 10 epochs for CIFAR-10, 20 epochs for CIFAR-100, and 5 epochs for ImageNet-1K. TDDS combines gradient information with training dynamics to estimate sample importance.

- **Bit-wise Unique Scores (Ours)**: Our method computes per-bit-width importance scores over 10 epochs for CIFAR-10, 20 epochs for CIFAR-100, and 5 epochs for ImageNet-1K, enabling dynamic coreset selection tailored to each sub-network.

## B  Additional Experiments

**Comparison of score uniqueness in bit-wise vs. batch-wise training scheme.** To analyze the effect of the proposed *bit-wise training scheme*, we examine the uniqueness of the importance scores it produces for each bit-width sub-model, compared to those from the standard *batch-wise training scheme*. Figure 7 shows a heatmap of Spearman correlation values, which quantify the similarity between importance scores produced by different bit-width sub-models. On both CIFAR-10 and CIFAR-100, the bit-wise scheme results in consistently lower correlation values, indicating that it yields more distinct and representative importance scores for each bit-width. This suggests that separating the training dynamics by bit-width is essential for accurately capturing per-sample

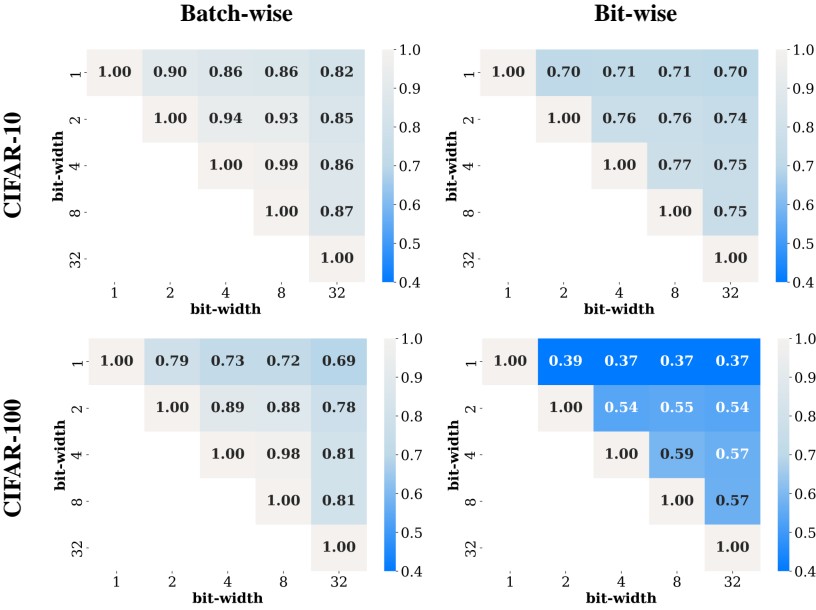

Figure 7: Comparison of score similarity across sub-models under batch-wise and bit-wise training schemes, measured by Spearman correlation. *The top row* shows PreActResNet-20 on CIFAR-10 results and *the bottom row* shows PreActResNet-18 on CIFAR-100; in each row, *the left* is batch-wise and *the right* is bit-wise.

importance in multi-bit settings, as it avoids the gradient aggregation seen in batch-wise training and allows each sub-model to maintain distinct context vectors. This isolation is critical for deriving meaningful, bit-wise importance estimates, which are otherwise masked under the standard batch-wise update pattern.

Table 8: PreActResNet-20 on CIFAR-10 and PreActResNet-18 on CIFAR-100. Breakdown of GPU hours.

| Dataset | Framework | GPU hours | | | | Total GPU hours (Speed up) |
|---|---|---|---|---|---|---|
| | | Training | Calibration | Adaptation | Scoring | |
| CIFAR-10 | Dedicated | 11.97 | - | - | - | 11.97 (1.00×) |
| | Any-Prec. | 7.51 | 1.25 | - | - | 8.76 (1.36×) |
| | **Bias Correction** | **7.51** | **-** | **0.004** | **-** | **7.52 (1.59×)** |
| | **Bias Correction + Coreset Sampling** | **1.52** | **-** | **0.004** | **0.37 (offline)** | **1.52 (7.88×)** |
| CIFAR-100 | Dedicated | 11.19 | - | - | - | 11.19 (1.00×) |
| | Any-Prec. | 7.17 | 1.10 | - | - | 8.27 (1.36×) |
| | **Bias Correction** | **7.17** | **-** | **0.004** | **-** | **7.17 (1.56×)** |
| | **Bias Correction + Coreset Sampling** | **1.47** | **-** | **0.004** | **0.74 (offline)** | **1.47 (7.61×)** |

**Breakdown of GPU hours.** We report GPU hours for all ResNet experiments, broken down by training stage, including: score evaluation, coreset training, calibration (if applicable), and adaptation. Table 8 also help clarify the computational cost structure of our framework. Specifically, coreset sampling significantly reduces training GPU hours for multi-bit quantization models, and this efficiency gain increases with the pruning rate. However, coreset sampling alone does not eliminate the cost of the calibration phase, which is typically needed to support additional bit-widths. To address this, we apply Bias Correction and BN adaptation, which allow us to remove the calibration step entirely without sacrificing accuracy.

**Experiments on alignment achieved by Bias Correction.** To demonstrate the effectiveness of our Bias Correction method, we conducted an analysis of the output activations, comparing how well activations from different bit-widths align with one another. In all experiments, the activation

Table 9: MAE between different bit-widths and full precision models' activation with vs. without Bias Correction.

| Bias Correction | MAE | | | | | | | |
|---|---|---|---|---|---|---|---|---|
| | 2bit | 3bit | 4bit | 5bit | 6bit | 7bit | 8bit | Avg. |
| - | 0.744 | 0.756 | 0.754 | 0.750 | 0.761 | 0.756 | 0.766 | 0.755 |
| ✔ | 0.667 | 0.670 | 0.671 | 0.654 | 0.683 | 0.668 | 0.659 | 0.667 |

precision is fixed to 4 bits, and we measure the mean absolute error (MAE) between the activations of quantized and full-precision models, with and without applying Bias Correction. Table 9 show that applying the correction consistently improves alignment, reducing the avg. MAE from 0.755 to 0.667. This improvement holds across all bit-widths, demonstrating our method's effectiveness. We will include these results in the revised version to highlight the impact of Bias Correction on activation alignment.

Table 10: PreActResNet-20 on CIFAR-10. BN sharing with and without 1-bit quantization.

| Framework | Test Accuracy | | | | | | | | | |
|---|---|---|---|---|---|---|---|---|---|---|
| | 1bit | 2bit | 3bit | 4bit | 5bit | 6bit | 7bit | 8bit | 32bit | Avg. |
| Share BN w/ 1bit | 46.38 | 91.90 | 92.57 | 91.48 | 90.98 | 90.53 | 90.24 | 90.14 | 89.90 | **86.01** |
| Share BN w/o 1bit | 92.49 | 89.44 | 92.43 | 92.67 | 92.43 | 92.15 | 92.00 | 91.97 | 91.65 | **91.91** |

**Experiments on effectiveness of BN sharing for 1-bit quantization.** In multi-bit settings, weight distributions tend to approximate a normal distribution, where our Bias Correction mechanism works effectively. However, in the 1-bit case, the weight distribution collapses into a near-uniform or binary form, causing a significant distribution shift that Bias Correction alone struggles to adequately address. For that reason, prior works on multi-bit networks often omit the 1-bit setting from their analysis [5, 18, 8]. In contrast, our method explicitly incorporates this case and demonstrates that assigning a separate BN layer for 1-bit quantization is both a simple and effective solution.

To further support this, we conduct ablation studies comparing two configurations: one where BN is shared across all bit-widths (including 1-bit), and another where the 1-bit case has its own BN, and the remaining bit-widths share a single BN. Table 10 clearly show that including 1-bit in BN sharing significantly degrades 1-bit quantization performance. On the other hand, using a separate BN for 1-bit achieves strong performance with negligible additional overhead, that is the additional parameters for the separate BN layer, which accounts for less than 0.01% of the total number of parameters. This performance degradation stems from unstable BN statistics. The 1-bit weights produce activation distributions that are markedly different from those of higher bit-widths, leading to significant fluctuations in shared BN statistics. These fluctuations negatively impact the normalization of other bit-widths, ultimately harming overall model performance.

Table 11: PreActResNet-20 on CIFAR-10. Impact of only coreset sampling.

| Coreset Sampling (Pruning Ratio) | Bias Correction | Test Accuracy | | | | | | | | | | GPU hours (Speed up) |
|---|---|---|---|---|---|---|---|---|---|---|---|---|
| | | 1bit | 2bit | 3bit | 4bit | 5bit | 6bit | 7bit | 8bit | 32bit | Avg. | |
| - (0.0) | - | 92.95 | 87.72 | 93.53 | 93.32 | 92.73 | 92.47 | 92.08 | 91.87 | 93.53 | 92.24 | 7.51 (**1.00×**) |
| ✔ (0.7) | - | 92.46 | 87.42 | 92.92 | 93.05 | 92.51 | 91.95 | 91.69 | 91.49 | 92.98 | 91.83 | 2.06 (**3.65×**) |
| | ✔ | 92.62 | 93.03 | 92.98 | 93.08 | 93.05 | 93.05 | 93.08 | 93.10 | 93.15 | 93.02 | |
| ✔ (0.8) | - | 92.29 | 87.61 | 92.84 | 92.27 | 91.51 | 90.99 | 90.68 | 90.46 | 92.84 | 91.28 | 1.52 (**4.94×**) |
| | ✔ | 92.60 | 93.01 | 92.96 | 93.03 | 93.02 | 92.99 | 93.01 | 93.00 | 93.08 | 92.97 | |
| ✔ (0.9) | - | 91.47 | 87.07 | 91.61 | 91.22 | 90.42 | 89.62 | 89.23 | 89.00 | 91.62 | 90.14 | 0.83 (**9.05×**) |
| | ✔ | 92.04 | 92.66 | 92.61 | 92.66 | 92.47 | 92.50 | 92.51 | 92.63 | 92.38 | 92.50 | |

**Experiments on impact of coreset sampling alone.** To examine the standalone contribution of coreset sampling to both accuracy and training speedup, we provide results on a CIFAR-10 baseline where only coreset sampling is applied, with all bit-widths except for 1-bit sharing BN layers. As shown in Table 11, while coreset sampling contributes most to the speedup, it is not sufficient on its

own to maintain strong accuracy at across every bit-width. In contrast, when Bias Correction and BN adaptation are applied alongside coreset sampling, we observe accuracy improvements everywhere, with pratically no additional GPU hours. The results indicate that although coreset sampling is the main driver of compute efficiency, Bias Correction and BN adaptation is essential for best accuracy.

Table 12: PreActResNet-20 on CIFAR-10. Influence of varying fixed temperatures and scheduling methods.

| Temperature scheme | | Test Accuracy | | | | | |
|---|---|---|---|---|---|---|---|
| | | 1bit | 2bit | 4bit | 8bit | 32bit | Avg |
| Fixed | 0.1 | 90.08 | 92.11 | 92.35 | 92.55 | 92.21 | **91.86** |
| | 0.5 | 90.61 | 92.61 | 92.78 | 92.85 | 92.61 | **92.29** |
| | 1.0 | 90.33 | 92.41 | 92.51 | 92.62 | 91.87 | **91.95** |
| Scheduling (0.1 ~ 1.0) | Linear | 90.40 | 92.29 | 92.07 | 92.22 | 91.55 | **91.71** |
| | Exp. | 90.57 | 92.39 | 92.56 | 92.63 | 91.93 | **92.02** |
| | Log | 90.77 | 92.64 | 92.65 | 92.89 | 92.06 | **92.20** |

**Experiments with varying fixed temperature settings and scheduling methods.** Table 12 presents the test accuracies of our method, evaluated in the Any-Precision setting, trained across bit-widths $b \in \{1, 2, 4, 8, 32\}$ using fixed sampling temperatures $\tau \in \{0.1, 0.5, 1.0\}$ and three temperature scheduling strategies (e.g., linear, exponential, logarithmic). We observe that among the fixed settings, a moderate temperature of $\tau = 0.5$ consistently achieves the highest accuracy, outperforming both the low ($\tau = 0.1$) and high ($\tau = 1.0$) extremes. Dynamically increasing the temperature from 0.1 to 1.0 over training—regardless of the scheduling scheme—yields performance that is comparable to or worse than using a fixed $\tau = 0.5$. These results suggest that a single, well-chosen temperature is sufficient to balance the sampling distribution—favoring informative samples while maintaining diversity. In contrast, dynamically adjusting the temperature throughout training introduces additional complexity without delivering clear performance benefits. Based on these observations, all coreset sampling experiments were conducted with the temperature fixed at $\tau = 0.5$.

Table 13: PreActResNet-20 on CIFAR-10 and PreActResNet–18 on CIFAR-100. Performance of calibrated bit-widths when pruning rate for *Coreset Sampling* is 80%.

| Dataset | Framework | Coreset Sampling | Test Accuracy | | | |
|---|---|---|---|---|---|---|
| | | | 3bit | 5bit | 6bit | 7bit |
| CIFAR-10 | Any-Prec. | - | 93.17 ±0.26 | 93.19 ±0.18 | 93.16 ±0.25 | 93.10 ±0.24 |
| | Ours | - | 93.46 ±0.14 | 93.54 ±0.02 | 93.50 ±0.07 | 93.43 ±0.10 |
| | | ✔ | 92.96 ±0.09 | 93.02 ±0.04 | 92.99 ±0.13 | 93.01 ±0.13 |
| CIFAR-100 | Any-Prec. | - | 71.28 ±0.26 | 71.43 ±0.24 | 71.53 ±0.15 | 71.45 ±0.09 |
| | Ours | - | 71.93 ±0.07 | 71.96 ±0.11 | 71.96 ±0.12 | 71.95 ±0.14 |
| | | ✔ | 70.41 ±0.15 | 70.42 ±0.10 | 70.47 ±0.08 | 70.57 ±0.13 |

**Evaluation of calibrated bit-widths.** In this section, we present the performance of calibrated bit-widths, which were omitted from the main experimental results in the main paper. Although we refer to these as *calibrated bit-widths*, it is important to clarify that, in our method, these bit-widths are not explicitly trained or fine-tuned. Instead, we obtain their accuracy using bias correction and batch normalization (BN) adaptation, without additional training or calibration stages. In contrast, Any-Precision [13] recovers calibrated bit-width performance by performing a separate post-training BN calibration procedure. As shown in Table 13 and Table 14, the calibrated bit-widths in our method achieve accuracy comparable to trained bit-widths, confirming that our proposed weight bias correction effectively aligns activation distributions without the need for costly calibration.

**Evaluation against baseline coreset selection methods on CIFAR-100.** Table 15 presents additional experimental results on CIFAR-100 using the PreActResNet-18 architecture, comparing our bit-wise coreset sampling method against several baseline coreset selection strategies. Consistent with prior findings in TDDS [17] and our CIFAR-10 experiments reported in the main paper, we observe that

Table 14: PreActResNet-20 on CIFAR-10 and PreActResNet-18 on CIFAR-100. Performance of calibrated bit-widths compared to previous methods at an 80% pruning rate.

| Dataset | Method | Test Accuracy | | | |
|---|---|---|---|---|---|
| | | 3bit | 5bit | 6bit | 7bit |
| CIFAR-10 | Random | 90.14 $_{\pm 0.25}$ | 90.15 $_{\pm 0.30}$ | 90.13 $_{\pm 0.32}$ | 90.10 $_{\pm 0.29}$ |
| | Entropy | 86.29 $_{\pm 0.37}$ | 86.30 $_{\pm 0.29}$ | 86.27 $_{\pm 0.27}$ | 86.28 $_{\pm 0.25}$ |
| | Forgetting | 78.12 $_{\pm 0.95}$ | 78.43 $_{\pm 1.01}$ | 78.37 $_{\pm 1.00}$ | 78.24 $_{\pm 1.02}$ |
| | EL2N | 81.15 $_{\pm 0.22}$ | 81.21 $_{\pm 0.38}$ | 81.20 $_{\pm 0.31}$ | 81.13 $_{\pm 0.43}$ |
| | Moderate | 88.36 $_{\pm 0.11}$ | 88.35 $_{\pm 0.15}$ | 88.39 $_{\pm 0.22}$ | 88.40 $_{\pm 0.19}$ |
| | TDDS | 88.72 $_{\pm 0.13}$ | 88.72 $_{\pm 0.07}$ | 88.70 $_{\pm 0.05}$ | 88.71 $_{\pm 0.01}$ |
| | **Ours** | 92.96 $_{\pm 0.09}$ | 93.02 $_{\pm 0.04}$ | 92.99 $_{\pm 0.13}$ | 93.01 $_{\pm 0.13}$ |
| CIFAR-100 | Random | 63.01 $_{\pm 0.50}$ | 63.00 $_{\pm 0.44}$ | 63.02 $_{\pm 0.42}$ | 63.09 $_{\pm 0.34}$ |
| | Entropy | 53.46 $_{\pm 0.31}$ | 53.45 $_{\pm 0.42}$ | 53.54 $_{\pm 0.51}$ | 53.53 $_{\pm 0.34}$ |
| | Forgetting | 38.71 $_{\pm 0.46}$ | 39.40 $_{\pm 0.55}$ | 39.49 $_{\pm 0.60}$ | 39.27 $_{\pm 0.72}$ |
| | EL2N | 31.00 $_{\pm 0.59}$ | 31.44 $_{\pm 0.62}$ | 31.57 $_{\pm 0.62}$ | 31.49 $_{\pm 0.55}$ |
| | Moderate | 58.54 $_{\pm 0.21}$ | 58.63 $_{\pm 0.19}$ | 58.67 $_{\pm 0.17}$ | 58.66 $_{\pm 0.13}$ |
| | TDDS | 54.40 $_{\pm 0.27}$ | 54.40 $_{\pm 0.40}$ | 54.40 $_{\pm 0.44}$ | 54.35 $_{\pm 0.40}$ |
| | **Ours** | 70.41 $_{\pm 0.15}$ | 70.42 $_{\pm 0.10}$ | 70.47 $_{\pm 0.08}$ | 70.57 $_{\pm 0.13}$ |

Table 15: PreActResNet-18 on CIFAR-100. Comparison with previous methods at 80% pruning rate.

| Method | Test Accuracy | | | | | |
|---|---|---|---|---|---|---|
| | 1bit | 2bit | 4bit | 8bit | 32bit | Avg. |
| Random | 60.32 $_{\pm 0.62}$ | 62.45 $_{\pm 0.51}$ | 63.11 $_{\pm 0.49}$ | 62.87 $_{\pm 0.53}$ | 61.39 $_{\pm 0.60}$ | 62.47 |
| Entropy | 52.65 $_{\pm 0.37}$ | 53.28 $_{\pm 0.20}$ | 53.55 $_{\pm 0.37}$ | 53.53 $_{\pm 0.47}$ | 53.22 $_{\pm 0.62}$ | 53.36 |
| Forgetting | 35.21 $_{\pm 0.37}$ | 38.00 $_{\pm 0.34}$ | 39.18 $_{\pm 0.50}$ | 39.45 $_{\pm 0.53}$ | 37.94 $_{\pm 0.68}$ | 38.52 |
| EL2N | 30.49 $_{\pm 0.91}$ | 31.07 $_{\pm 0.57}$ | 31.38 $_{\pm 0.62}$ | 31.50 $_{\pm 0.57}$ | 29.43 $_{\pm 0.29}$ | 31.04 |
| Moderate | 57.05 $_{\pm 0.40}$ | 58.24 $_{\pm 0.42}$ | 58.69 $_{\pm 0.16}$ | 58.67 $_{\pm 0.10}$ | 57.83 $_{\pm 0.37}$ | 58.33 |
| TDDS | 53.65 $_{\pm 0.36}$ | 54.07 $_{\pm 0.43}$ | 54.40 $_{\pm 0.40}$ | 54.36 $_{\pm 0.41}$ | 54.04 $_{\pm 0.55}$ | 54.23 |
| **Ours** | 69.14 $_{\pm 0.08}$ | 70.12 $_{\pm 0.11}$ | 70.35 $_{\pm 0.17}$ | 70.43 $_{\pm 0.11}$ | 70.41 $_{\pm 0.11}$ | **70.26** |

random coreset selection performs surprisingly well at high pruning rates. This trend persists in the CIFAR-100 setting, where random sampling remains a strong baseline under severe data reduction. Nonetheless, our proposed method consistently outperforms the baselines across different bit-widths, demonstrating its effectiveness in selecting informative samples even under high pruning constraints.

**Evaluation of DeiT-S on CIFAR-100 and TinyImageNet** We further evaluate our method on a transformer-based architecture, specifically DeiT-S, using smaller datasets such as CIFAR-100 and TinyImageNet. The results are presented in Table 16. To the best of our knowledge, there is currently no standardized multi-bit framework like Any-Precision for vision transformers. To address this, we implement our own framework following configurations similar to Any-Precision, with a slight modification to the StatsQ quantizer—details of which are provided in Section A. Our method demonstrates consistently strong performance compared to the basic Any-Precision setup, even when pruning the dataset by 60%, achieving up to an $8.41\times$ reduction in GPU hours on TinyImageNet.

**Evaluation of storage-constrained coreset sampling.** Coreset-based approaches in multi-bit networks consistently reduce training time; however, due to varying data importance across sub-models, it remains challenging to impose a uniform constraint on the total number of training samples used by the entire model. To address this, we first discard samples that are consistently considered unimportant across all sub-models, and then apply our coreset sampling method with bias correction. To identify and remove consistently uninformative samples before applying coreset sampling, we first compute the importance of each training sample for every sub-model over a single epoch, following our bit-wise training scheme. These importance values are then summed across sub-models, and their variability is assessed—similar to training dynamics approaches [17, 16]—to obtain a

Table 16: DeiT-S on CIFAR-100 and TinyImageNet.

| Dataset | Framework | Pruning Rate | Test Accuracy | | | | | GPU hours (Speed up) |
|---|---|---|---|---|---|---|---|---|
| | | | 2bit | 4bit | 6bit | 8bit | Avg. | |
| CIFAR-100 | Dedicated | - | 87.14 | 87.92 | 87.88 | 88.03 | 87.74 | 41.01 (1.00×) |
| | Any-Prec. | - | 87.52 | 88.30 | 88.20 | 88.21 | 88.08 | 10.47 (3.92×) |
| | Ours | 50% | 87.83 | 88.56 | 88.68 | 88.59 | **88.43** | 6.05 (**6.78×**) |
| | | 60% | 87.61 | 88.45 | 88.54 | 88.65 | **88.31** | 5.20 (**7.89×**) |
| TinyImageNet | Dedicated | - | 82.61 | 85.60 | 85.68 | 85.86 | 84.94 | 74.00 (1.00×) |
| | Any-Prec. | - | 82.10 | 84.61 | 84.47 | 84.70 | 84.07 | 19.32 (3.83×) |
| | Ours | 50% | 82.54 | 84.95 | 85.33 | 85.17 | **84.59** | 10.50 (**7.05×**) |
| | | 60% | 82.89 | 84.39 | 84.95 | 84.86 | **84.26** | 8.80 (**8.41×**) |

Table 17: PreActResNet-20 on CIFAR-10. Comparison across varying storage reduction rates.

| Storage Reduction | Test Accuracy | | | | | |
|---|---|---|---|---|---|---|
| | 1bit | 2bit | 4bit | 8bit | 32bit | Avg. |
| 0% | 92.60 | 93.01 | 93.03 | 93.00 | 93.08 | **92.97** |
| 20% | 92.39 | 93.17 | 93.26 | 93.32 | 93.32 | **93.20** |
| 30% | 92.28 | 92.70 | 92.68 | 92.62 | 93.10 | **92.65** |
| 40% | 92.09 | 92.61 | 92.75 | 92.77 | 92.55 | **92.64** |
| 50% | 91.97 | 92.15 | 92.19 | 92.29 | 92.61 | **92.25** |

final importance score. As shown in Table 17, our coreset sampling method performs comparably to existing approaches, even under a 50% dataset storage constraint. Moreover, Table 18 shows that training performance can be further enhanced by tuning the pruning rate (i.e., training time), highlighting the adaptability of our method to varying resource budgets in multi-bit network training.

**Experiments on Influence of coreset sampling frequency.** In practice, the overhead of bit-wise coreset resampling is extremely small compared to the overall training cost. For example, even on an ImageNet-scale dataset, performing 100 resamplings takes only about 3.36 minutes. Given this negligible cost, resampling at every epoch is a practical and effective choice.

To quantitatively demonstrate this, we conducted experiments with different resampling intervals and measured both validation accuracy and total sampling time. Table 19 show that resampling every epoch improves average accuracy by 1.33%p compared to resampling every 30 epochs, while adding just 53 seconds of overhead to a multi-hour training process. This demonstrates that frequent resampling can offer meaningful accuracy gains at virtually no additional cost.

**Experiments on dynamic score re-evaluation.** We conducted additional experiments where importance scores are dynamically re-evaluated every 10, 30, 50, or 100 epochs, and coresets are resampled accordingly. We evaluated how it impacts accuracy and GPU hours using PreActResNet-20 on CIFAR-10 and PreActResNet-18 on CIFAR-100 under both 80% and 90% data pruning. Table 20 and Table 21 reveal a consistent pattern: while dynamic score re-evaluation leads to only marginal accuracy changes, it incurs a substantial increase in computational cost. In many settings, our one-time scoring strategy already matches or even outperforms more frequent re-evaluation in terms of final accuracy, while consuming significantly fewer GPU hours. This empirical finding validates our design choice, and shows that a single, well-computed importance estimate—when paired with stochastic sampling—offers an effective and efficient balance, capturing most of the benefits of dynamic importance tracking without incurring its heavy cost. Looking ahead, with the observation that dynamic re-evaluation yields modest gains on the more challenging CIFAR-100, we believe dynamic sampling techniques could be the key to boosting performance on complex, high-variability

Table 18: PreActResNet-20 on CIFAR-10 and PreActResNet-18 on CIFAR-100. Comparison across pruning rates when dataset storage is fixed at 30K out of 50K samples. Since retaining 30K out of 50K samples represents a 40% reduction, a pruning rate of 40% corresponds to the full-training scenario in this context.

| Dataset | Pruning Rate | Test Accuracy | | | | | |
|---|---|---|---|---|---|---|---|
| | | 1bit | 2bit | 4bit | 8bit | 32bit | Avg. |
| CIFAR-10 | 40% (full) | 93.01 | 93.39 | 93.61 | 93.58 | 93.65 | **93.50** |
| | 50% | 92.81 | 93.07 | 93.11 | 93.15 | 93.39 | **93.13** |
| | 60% | 92.66 | 93.07 | 93.04 | 93.09 | 93.48 | **93.07** |
| | 70% | 92.52 | 92.91 | 93.07 | 93.11 | 93.17 | **92.97** |
| | 80% | 92.09 | 92.61 | 92.75 | 92.77 | 92.55 | **92.64** |
| CIFAR-100 | 40% (full) | 66.01 | 66.35 | 66.85 | 66.90 | 66.90 | **66.65** |
| | 50% | 66.32 | 66.98 | 67.60 | 67.93 | 67.49 | **67.52** |
| | 60% | 65.65 | 65.99 | 66.48 | 66.38 | 66.20 | **66.24** |
| | 70% | 66.10 | 66.86 | 66.69 | 66.90 | 67.03 | **66.78** |
| | 80% | 65.29 | 66.30 | 66.33 | 66.39 | 65.83 | **66.19** |

Table 19: PreActResNet-20 on CIFAR-10. Influence of sampling frequency.

| Resampling Frequency | Test Accuracy | | | | | | Total Sampling Time (% of Total GPU time) |
|---|---|---|---|---|---|---|---|
| | 1bit | 2bit | 4bit | 8bit | 32bit | Avg. | |
| 1 | 92.60 | 93.01 | 93.03 | 93.00 | 93.08 | **92.97** | 53.02s  **(0.96%)** |
| 10 | 91.96 | 92.40 | 92.65 | 92.66 | 92.75 | **92.55** | 5.09s  **(0.09%)** |
| 20 | 91.40 | 91.73 | 92.04 | 92.03 | 92.34 | **91.98** | 2.77s  **(0.05%)** |
| 30 | 91.29 | 91.53 | 91.63 | 91.73 | 91.68 | **91.64** | 1.40s  **(0.03%)** |

datasets where importance scores drift more drastically throughout training. The main hurdle is the high cost of score re-evaluation during training, which currently limits the practicality of dynamic methods. In future work, we will explore lightweight techniques to reduce score-evaluation overhead while maintaining the quality of importance estimates.

## C   Theoretical Analysis of Cross-bit-width Implicit Knowledge Transfer

In this section, we use a simple linear classifier to examine how shared weights in multi-bit networks can *implicitly* transfer knowledge between sub-networks. We consider a setting where an 8-bit model and a 2-bit model share the same real-valued parameter vector $w$, with weights quantized using the DoReFa quantizer [41]. Training alternates iteratively: the 8-bit model is trained on batch $X_8$ with hard labels, followed by the 2-bit model trained on a separate batch $X_2$, also with hard labels. These batches are drawn independently from the data matrix $X \in \mathbb{R}^{d \times n}$ and do not overlap. The shared parameter $w$ is updated in-place using the cross-entropy loss and is continuously modified by both models. The key question is: *can we formally argue that, despite no explicit soft-label distillation and no shared data examples, the 2-bit model benefits from the 8-bit model's training- and vice versa-through the shared parameter?*

**Gradient update within combined data subspaces.** When the 8-bit model observes batch $B_8 = (X_8, y_8)$, it performs a gradient step using the cross-entropy loss. The model is blind to any component of the optimum that is orthogonal to the plane that spans the $n_8$-dimensional subspace of $X_8$ [45]. That is, under an asymptotic assumption, the direction of the update is fully constrained to the subspace spanned by the input vectors in the batch, as the gradient is a linear combination of the each data point $x_i$. Therefore, the gradient of the 8-bit model lies within the subspace spanned by its input batch, and thus the corresponding update step is bound as follows: $\Delta_8 \in \text{span}(X_8)$. Similarly, the 2-bit model performs its update with batch $X_2$. By induction, the net update to $w$ lies within a sum

Table 20: PreActResNet-20 on CIFAR-10. Impact of score re-evaluation.

| Pruning Ratio | Re-eval Frequency | Test Accuracy | | | | | | Re-eval GPU hours | Total GPU hours |
|---|---|---|---|---|---|---|---|---|---|
| | | 1bit | 2bit | 4bit | 8bit | 32bit | Avg. | | |
| 0.8 | - | 92.60 | 93.01 | 93.03 | 93.00 | 93.08 | **92.97** | - | **1.52** |
| | 100 | 92.08 | 92.64 | 92.80 | 92.82 | 92.88 | **92.75** | 0.38 | **5.31** |
| | 50 | 92.45 | 92.83 | 92.96 | 93.04 | 92.96 | **92.97** | 1.13 | **2.65** |
| | 30 | 92.64 | 92.90 | 93.03 | 92.88 | 92.69 | **92.84** | 2.27 | **3.79** |
| | 10 | 92.21 | 92.56 | 92.73 | 92.86 | 92.75 | **92.73** | 7.33 | **8.85** |
| 0.9 | - | 92.04 | 92.66 | 92.66 | 92.63 | 92.38 | **92.38** | - | **0.84** |
| | 100 | 91.43 | 92.09 | 92.09 | 92.09 | 91.89 | **92.09** | 0.37 | **1.21** |
| | 50 | 91.37 | 91.98 | 92.22 | 92.29 | 92.32 | **92.09** | 1.15 | **1.99** |
| | 30 | 91.94 | 92.13 | 92.12 | 92.11 | 92.38 | **92.10** | 2.31 | **3.15** |
| | 10 | 91.24 | 91.64 | 91.94 | 92.03 | 91.77 | **91.82** | 7.33 | **8.17** |

Table 21: PreActResNet-18 on CIFAR-100. Impact of score re-evaluation.

| Pruning Ratio | Re-eval Frequency | Test Accuracy | | | | | | Re-eval GPU hours | Total GPU hours |
|---|---|---|---|---|---|---|---|---|---|
| | | 1bit | 2bit | 4bit | 8bit | 32bit | Avg. | | |
| 0.8 | - | 69.14 | 70.12 | 70.35 | 70.43 | 70.41 | **70.26** | - | **1.47** |
| | 100 | 69.16 | 70.24 | 70.41 | 70.39 | 70.04 | **70.18** | 0.77 | **2.24** |
| | 50 | 69.13 | 70.23 | 70.60 | 70.82 | 70.41 | **70.45** | 2.30 | **3.78** |
| | 30 | 68.91 | 70.17 | 70.18 | 70.05 | 70.09 | **70.02** | 4.57 | **6.04** |
| | 10 | 68.95 | 70.45 | 70.76 | 70.78 | 70.87 | **70.66** | 14.57 | **16.04** |
| 0.9 | - | 67.53 | 69.40 | 69.75 | 69.25 | 69.24 | **69.32** | - | **0.83** |
| | 100 | 67.34 | 69.17 | 69.23 | 69.29 | 69.28 | **69.02** | 0.76 | **1.59** |
| | 50 | 67.77 | 69.12 | 69.43 | 69.20 | 69.11 | **69.11** | 2.32 | **3.15** |
| | 30 | 67.82 | 69.30 | 69.11 | 69.34 | 69.02 | **69.15** | 4.63 | **5.46** |
| | 10 | 67.28 | 69.14 | 69.41 | 69.50 | 69.44 | **69.23** | 14.68 | **15.51** |

of subspaces as follows: $\Delta_{\text{net}} \in \sum_{j=1}^{\mathcal{B}} \text{span}(X_j)$, where $\mathcal{B} = \{2, 8\}$ in our case. Thus, the shared weight vector evolves within the union of data subspaces: $\text{span}(X_8) \cup \text{span}(X_2)$. This shows that each sub-network updates its parameters based on a broader subspace that includes data from other sub-networks, thereby increasing its effective data exposure.

**Gradient alignment between quantized sub-networks.** Given that updates occur within a shared subspace, we analyze whether the gradients from different quantized sub-networks are sufficiently aligned to enable mutual benefit. We assume that the optimum value for both quantized model is similar [41]. In our setting, when the 8-bit model receives batch $X_8$, this updates $w$ towards minimizing $\mathcal{L}_8$. Since $\Delta w_8 \in \text{span}(X_8)$, this update steers $w$ toward the optimum $w^\star$ within $\text{span}(X_8)$. The 8-bit model moves $w$ to a point where the 2-bit loss cannot be worse—and is often better (i.e., $\theta < 90$). This is functionally equivalent to soft distillation: the 8-bit model's higher-capacity updates are immediately used by the 2-bit model, enabling generalization benefits without soft targets. This provides a theoretical basis for implicit knowledge transfer as the shared parameter acts as a channel of indirect supervision.

Motivated by these observations, we propose a bit-wise coreset sampling method that trains each sub-network with its own compact, informative subset of the full dataset. As the multi-bit network implicitly accesses the collective data seen by all sub-networks, each can prioritize important samples by directly training them, while also benefiting from diverse data exposure through indirect supervision. This not only reduces the overall training cost but also preserves model performance by ensuring sufficient coverage of the dataset across bit-widths.

