# OpenReview forum: "Efficient Multi-bit Quantization Network Training via Weight Bias Correction and Bit-wise Coreset Sampling"
_NeurIPS.cc/2025/Conference — NeurIPS 2025 poster_

### Official Review · Reviewer_RkBj · 2025-07-01

**Clarity:** 3
**Significance:** 2
**Originality:** 3
**Rating:** 4
**Confidence:** 3

**Summary:**

They proposes two techniques to reduce the training overhead of multi-bit quantization networks. The first technique is weight bias correction, which aligns the quantized weights to match the activation outputs across different bit-widths, enabling shared batch normalization and eliminating the need for an extra training stage. The second technique is bit-wise coreset sampling, which dynamically selects informative data subsets for each child model based on gradient-based importance scores, reducing training redundancy. Experiments on CIFAR-10/100, TinyImageNet, and ImageNet-1K with ResNet and ViT architectures demonstrate that the proposed methods achieve competitive or superior accuracy while significantly reducing training time.

**Questions:**

See Strengths And Weaknesses

**Ethical Concerns:**

["NO or VERY MINOR ethics concerns only"]

**Final Justification:**

I appreciate author's detailed rebuttal to address my concerns. I have read the author response and decide to maintain my recommendation.

**Limitations:**

See Strengths And Weaknesses

**Quality:**

3

**Strengths And Weaknesses:**

Strengths
1. The paper introduces two novel techniques, which are well-motivated and supported by both theoretical insights and empirical results. It is worth noting that this method is applicable to both ViT and CNNs and is not restricted by the BN layer.
2. The proposed methods achieve up to 7.88× reduction in GPU hours, making multi-bit quantization networks more practical for deployment.
3. The authors conduct extensive experiments across multiple datasets and architectures, demonstrating the effectiveness and generalizability of their approach.
4. The paper is well-organized and clearly written. The authors provide detailed explanations of their methods, supported by intuitive figures and comprehensive tables, making it easy for readers to understand and reproduce the work.

Weaknesses
1. The evaluations are restricted to computer vision tasks. While this is understandable given the computational cost of training multi-bit networks, extending the approach to other domains such as generative LLMs would further validate the broad applicability of the methods. Of course, this is merely encouragement rather than a requirement.
2. Eq. 2 was not properly introduced and was somewhat abrupt. The paper does not provide a formal theoretical analysis or explanation of the proposed techniques of Eq. 2. Why Eq.2 reduce the errors?
3. Furthermore, does the [29] mentioned in line 165 have been utilized in the ViT network? This might need some ablation experiments.
4. Implementing the bit-wise coreset sampling strategy might be complex and require significant modifications to existing training pipelines. The authors could provide more detailed guidance or tools to facilitate adoption by the research community.

---

> ### Author Rebuttal · Authors · 2025-07-31
>
> We would like to extend our heartfelt appreciation to the reviewer for your careful and constructive feedback, which has profoundly enriched the quality and clarity of our work.
>
> **W1: Broader validation on other domains would strengthen applicability.**
>
> We fully agree that validating our approach on other domains such as generative LLMs would further strengthen its general applicability and impact. However, exploring multi-bit quantization in the context of LLMs is currently not technically feasible for us, primarily due to the substantial computational resources required to train such models from scratch. Moreover, this application domain remains relatively understudied, and additional foundational studies are needed to understand the behavior of multi-bit quantization in LLM settings. A further challenge is that existing sample-importance scoring methods are tailored to discriminative models, and adapting them for generative models remains an open problem. We therefore consider this an important direction for future work and will include a discussion of this in the revised manuscript.
>
> **W2: Provide the theoretical basis behind Eq. 2 and how it contributes to error reduction.**
>
> Thank you for pointing this out. We agree that Eq. 2 was introduced somewhat abruptly, and hope to clarify the theoretical groundings and provide a more detailed explanation of how it contributes to reducing activation errors below.
>
> Our goal is to mitigate the mismatch in activation distributions across bit-widths by addressing the root cause: systematic bias introduced during weight quantization. Since we isolate the effect of quantized weights by fixing the input activation, we analyze the convolution output as follows. Let:
> - $x \in \mathbb{R}^d$: fixed input activation (same across all bit-widths),
> - $w, w_q \in \mathbb{R}^d$: full-precision and quantized weight vectors,
> - $y = x^\top w$: full-precision output activation,
> - $y_q = x^\top w_q$: quantized output activation.
>
> Then the error induced by quantization is: $\epsilon = y_q - y = x^\top(w_q - w)$. This shows that any discrepancy in output activations is a direct result of the quantization error in the weight vector, scaled by the input activation. Since $x$ is fixed, the variation across bit-widths in $y_q$ comes solely from $w_q$. As shown in Fig. 3, quantized weights ($w_q$) tend to introduce systematic shifts in both mean and scale compared to the full-precision weights ($w$), leading to:
> - $\mathbb{E}[w_q] \neq \mathbb{E}[w]$ (bias mismatch),
> - $\mathbb{V}[w_q] \neq \mathbb{V}[w]$ (variance mismatch).
>
> These mismatches result in distributional shifts in the output activation $y_q$, which become more severe under lower bit-widths. Our proposed correction aims to eliminate these mismatches at the source, i.e., the weight level. We seek a transformation of $w_q$ that makes it aligned with $w$ in terms of mean and variance. Let’s define a corrected quantized weight vector $w_q’$ such that:
> 	1.	$\mathbb{E}[w_q’] = \mathbb{E}[w]$
> 	2.	$\mathbb{V}[w_q’] = \mathbb{V}[w]$
>
> To align the quantized weights with the full-precision weights, we apply an affine transformation of the form $w_q’ = \alpha (w_q + \beta)$, where $\alpha \in \mathbb{R}$ controls scaling and $\beta \in \mathbb{R}^d$ controls shifting. To match the mean, we require $\mathbb{E}[w_q’] = \alpha (\mathbb{E}[w_q] + \beta) = \mathbb{E}[w]$, which gives $\beta = \mathbb{E}[w] - \mathbb{E}[w_q]$. To match the variance, we require $\mathbb{V}[w_q’] = \alpha^2 \mathbb{V}[w_q] = \mathbb{V}[w]$, which yields $\alpha = \sqrt{ \frac{\mathbb{V}[w]}{\mathbb{V}[w_q]} }$. Substitute $\alpha$ and $\beta$ into the affine form, and we get the Bias Correction equation:
> $$
> w_q’ = \sqrt{ \frac{\mathbb{V}[w]}{\mathbb{V}[w_q]} } \left(w_q + \left(\mathbb{E}[w] - \mathbb{E}[w_q]\right)\right).
> $$
>
> Using this corrected $w_q’$, the new activation becomes:
> $$
> y_q’ = x^\top \left( \sqrt{ \frac{\mathbb{V}[w]}{\mathbb{V}[w_q]} } (w_q + \mathbb{E}[w] - \mathbb{E}[w_q]) \right).
> $$
>
> The expression above can be interpreted as an affine transformation of $y_q$ (i.e., rescaled and shifted), controlled by the quantized weight distribution’s discrepancy from the full-precision distribution. Since the input $x$ is fixed, correcting the weight vector’s mean and variance directly adjusts the distribution of the output $y_q’$, reducing systematic activation shifts and scale mismatches across bit-widths. In doing so, the challenge of aligning activation distributions is effectively reduced to a simpler and more stable problem of aligning weight statistics. As a result, different quantized sub-networks can share BN layers more reliably, as the activations are already well-aligned at the source (i.e., before normalization), eliminating the need for separate BN parameters or additional activation calibration. We will incorporate this detailed theoretical grounding into the main manuscript and provide the full derivation and extended discussion in an appendix of the revised paper.
>
> **W3: Clarify whether BN adaptation has been applied to ViTs.**
>
> ViTs use Layer Normalization (LN) instead of Batch Normalization (BN), eliminating the need for BN-specific calibration, such as updating running mean and variance after training. Unlike BN, LN does not maintain such running statistics. As far as we are aware, there has been no prior investigation into how LN behaves in the context of multi-bit ViT networks. Our work is the first to experimentally demonstrate that LN parameters, once trained, remains relatively effective across all bit-widths without requiring additional calibration or adjustment. We will incorporate this discussion and the detailed results into the Appendix
>
> **W4: Implementation of bit-wise coreset sampling into existing pipelines seems complex, provide more guidance.**
>
> We understand that ease of implementation is crucial for broader adoption, and we truly appreciate the feedback. In practice, our scoring and sampling code is already modularized, making it easy to plug into various network architectures and base models. To further support integration, we will provide a detailed guide covering key components such as pruning rate $p$, sampling frequency, and the temperature variable—three of the few hyperparameters closely tied to our method. We also plan to automate the full pipeline with a single bash script and include clear instructions for each stage (scoring, sampling, training, and if applicable, BN adaptation). Finally, we will publicly release all code and scripts to facilitate easy adoption and reproducibility.

---

> > ### Comment · Reviewer_RkBj · 2025-08-05
> >
> > I appreciate author's detailed rebuttal to address my concerns. I have read the author response and decide to maintain my recommendation.

---

### Official Review · Reviewer_8UF3 · 2025-07-01

**Clarity:** 3
**Significance:** 3
**Originality:** 2
**Rating:** 4
**Confidence:** 4

**Summary:**

This paper proposes a method to reduce the computation required to train multi-bit quantization models. The authors observed that the quantization error in the model comes from a bias between the different quantization widths and propose to address this bias by directly modifying the weights. This results in a lighter model that does not need a batch normalization layer per quantization width. Additionally, the authors also propose an efficient strategy to calibrate the model that leverages core set selection to reduce the number of samples that the model needs to visit, further reducing the computation required to train the model.

**Questions:**

NA

**Ethical Concerns:**

["NO or VERY MINOR ethics concerns only"]

**Final Justification:**

The authors addressed most of the reviewers' questions.

**Quality:**

3

**Strengths And Weaknesses:**

Strengths:
- The paper is well written and easy to follow. The visualizations effectively support the claims in the text and help understand the proposed approach.
- The approach is well designed: each element tackles a weakness of the task in an efficient manner. As a result, the proposed approach surpasses SOTA approaches in accuracy and computation efficiency.
- The experimental section is comprehensive and well-designed.


Weaknesses:
- The novelty of the proposed approach is limited, and the design is somewhat simple. The contribution of the paper could be stronger with a more detailed discussion or a theoretical framework that explained the effect of the different elements of the model. This would bring valuable insights to the community.
- The efficiency benefits of the coreset sampling strategy should be better highlighted. This is an important contribution, and there is no formal definition of the computation required in relation to a standard approach.

---

> ### Author Rebuttal · Authors · 2025-07-31
>
> We sincerely appreciate the reviewer’s thoughtful and detailed feedback, which has greatly enhanced the depth, clarity, and overall quality of our work.
>
> **W1: Strengthen the contribution with more detailed discussion or theoretical insights.**
>
> We appreciate the suggestion and agree that a deeper theoretical discussion can greatly enhance the contribution. In response, we would like to highlight *Section C* of our appendix, which provides a detailed theoretical analysis of cross-bit-width implicit knowledge transfer—a core aspect of our method. Specifically, we offer a formal perspective on how sub-networks with different quantization levels (e.g., 2-bit and 8-bit) can implicitly benefit from each other by sharing weights, even without overlapping data or explicit soft-label supervision. We show that the gradient updates from each sub-network lie within the span of their respective input data subspaces, and that the shared parameter is effectively updated over the union of these subspaces. This results in an implicit data-sharing effect that broadens the effective training distribution for each sub-network (Lines 143–155). Furthermore, we theoretically analyze the alignment between gradients from differently quantized sub-networks and argue that this serves as a form of indirect supervision—akin to soft distillation—without requiring additional targets or training overhead (Lines 156–165). This insight is, to the best of our knowledge, unique to our work and has not been previously formalized in prior studies on multi-bit quantization.
>
> We believe this framework not only strengthens the novelty of our method, but also provides valuable insights into the design principles behind multi-bit quantized networks trained with shared weights. We will make sure to better highlight these points in the main text for clarity.
>
> **W2: More clearly highlight the efficiency gains of coreset sampling with a formal comparison to standard approaches.**
>
> We appreciate your helpful feedback. To more clearly highlight the efficiency gains of our proposed coreset sampling strategy, we have added a formal comparison to standard approches (i.e., Any-Precision) in the Table below. This breakdown presents the GPU hours required for training, adaption, calibration, and scoring steps under different frameworks using ResNet-20 on CIFAR-10 with pruning rate of 80%. We excluded the scoring time from GPU hours, as it is an offline process and can be reused across all experiments under the same or comparable experimental settings (e.g., dataset, model architecture).
>
> **[Table. Breakdown of GPU hours for ResNet-20 experiments with 80% pruning]**
> |Framework|Training (Speedup)|Calibration|Adaptation|Scoring|Total GPU hours (Speedup)|
> |-|-|-|-|-|-|
> |Any-Prec. (Standard)|7.51 (1.00×)|1.25|-|-|8.76 (1.00×)|
> |Bias Correction (Ours)|7.51 (1.00×)|-|0.004|-|7.52 (1.16×)|
> |Bias Correction+Coreset Sampling (Ours)|1.52 **(4.94×)**|-|0.004|0.37 *(Offline)*|1.52 **(5.76×)**|
>
> With these results, we would like to briefly clarify our framework. First, coreset sampling significantly reduces the training GPU hours for multi-bit quantization models. This efficiency gain grows in proportion to the pruning rate. However, coreset sampling on its own does not remove the cost of the calibration phase, which is required to support additional bit-widths. To address this, we apply Bias Correction and BN adaptation, effectively eliminating the need for calibration. Therefore, if we formally define the computational cost of each approach with the pruning rate $p \in [0,1)$, where $p$ is the fraction of samples removed from the full dataset, it can be expressed as:
>
> Cost of the Standard Approach (Any-Precision):
> $$
> \text{Full-training time per bit} \times \text{The number of trained bit-widths} + \text{Calibration time per bit} \times \text{The number of calibrated bit-widths}
> $$
>
> Cost of Our Method (Bias Correction + Coreset Sampling):
> $$
> (1 - p) \times \text{Full-training time per bit} \times \text{The number of trained bit-widths}
> $$
>
> Here, calibrated bit-widths refer to intermediate bit-widths that are not directly trained but require calibration, such as 3, 5, 6 and 7 bits. BN adaptation incurs only a negligible overhead of 0.004 GPU hours as shown in the table, so we omit it from the formula for clarity.
>
> We hope this clarification, along with the updated comparison table, clearly demonstrates the efficiency benefits of our framework. By jointly applying coreset sampling, Bias Correction, and BN adaptation, our method achieves substantial reductions in computational cost while preserving accuracy across bit-widths.

---

> > ### Comment · Reviewer_8UF3 · 2025-08-05
> >
> > Thank you to the authors for addressing all my concerns. After reading the other reviewers' comments and the corresponding responses, my score remains the same.

---

### Official Review · Reviewer_Txhn · 2025-07-01

**Clarity:** 2
**Significance:** 3
**Originality:** 3
**Rating:** 4
**Confidence:** 4

**Summary:**

This work proposes an efficient training framework for producing multi-bit neural networks. The authors introduce a weight bias correction technique to align activation distributions, enabling the use of shared batch normalization during training and shifting the calibration burden to a more simple post-training adaptation step. Additionally, they propose a novel bit-wise dynamic coreset sampling method to significantly reduce training time. Empirical validation and comparisons with existing methods demonstrate the effectiveness of the proposed approach.

**Questions:**

- Regarding weakness1, could the authors clarify this design choice and provide result of applying dynamic coreset sampling on spearman correlation?
- Could the authors elaborate concerns on weakness 2 and 3?
- Could the authors provide GPU hours with more detailed granularity— the time spent separately on training, calibration, and adaptation phases? This would help better assess the efficiency claims of the proposed method.

**Ethical Concerns:**

["NO or VERY MINOR ethics concerns only"]

**Final Justification:**

Authors have addressed all of my concerns, and I raise my score

**Limitations:**

Limitations and potential societal impact are not addressed in the paper

**Paper Formatting Concerns:**

.

**Quality:**

3

**Strengths And Weaknesses:**

Strengths:
1. Building on observations, the authors propose a novel approach to handle activation misalignment and introduce coreset sampling dedicated for multi-bit training.
2. The method demonstrates significant improvements in wall-clock training time while maintaining quality, validating its effectiveness.


Weaknesses:
1. If my understanding is correct, sample-wise importance changes throughout the training process. However, the proposed sampling strategy calculates importance only once, prior to coreset selection, without updating it over time. Even though temperature-based sampling is used, samples with initially high probability are more likely to be selected consistently, which seems misaligned with the motivation of dynamically capturing changing important samples.
2. The proposed weight bias correction is claimed to enable shared batch normalization during training. However, a separate BN layer is still used for the 1-bit setting. Does this imply that the bias correction mechanism is less effective or insufficient for 1-bit quantization?
3. In the ImageNet-1K experiments, the paper mentions that the Any-Precision model is used for fine-tuning. Does this mean that fine-tuning is performed starting from a pre-trained Any-Precision model after the calibration or training phase? Clarification on the training pipeline would be appreciated.

---

> ### Author Rebuttal · Authors · 2025-07-31
>
> Thank you for your thoughtful feedback and insightful suggestions. In response, we have carefully followed your recommendations and provide additional results and clarifications below.
>
> **W1: Dynamic sample-wise importance**
>
> We appreciate the reviewer’s insightful question. You are absolutely right that accounting for the dynamic nature of sample importance is a key consideration of our method. Our decision to compute importance scores only once at the beginning—rather than periodically throughout training—was primarily driven by the high computational cost associated with full-dataset scoring. To address the potential rigidity of this one-time evaluation, we incorporate temperature-based probabilistic sampling, which introduces stochasticity into the coreset selection process. This allows our method to be aware of potential importance variations, even without explicitly updating the scores over time.
>
> That said, we acknowledge the validity of the reviewer’s concern that high-probability samples may still dominate selection, especially when importance is fixed. To directly examine this, we conducted additional experiments where importance scores are dynamically re-evaluated every 10, 30, 50, or 100 epochs, and coresets are resampled accordingly. We evaluated how it impacts accuracy and GPU hours using ResNet-20 on CIFAR-10 and ResNet-18 on CIFAR-100 under both 80% and 90% data pruning.
>
> The results below reveal a consistent pattern: while dynamic score re-evaluation leads to only marginal accuracy changes, it incurs a substantial increase in computational cost. In many settings, our one-time scoring strategy already matches or even outperforms more frequent re-evaluation in terms of final accuracy, while consuming significantly fewer GPU hours. This empirical finding validates our design choice, and shows that a single, well-computed importance estimate—when paired with stochastic sampling—offers an effective and efficient balance, capturing most of the benefits of dynamic importance tracking without incurring its heavy cost.
>
> Looking ahead, with the observation that dynamic re-evaluation yields modest gains on the more challenging CIFAR-100, we believe dynamic sampling techniques could be the key to boosting performance on complex, high-variability datasets where importance scores drift more drastically throughout training. The main hurdle is the high cost of score re-evaluation during training, which currently limits the practicality of dynamic methods. In future work, we will explore lightweight techniques to reduce score-evaluation overhead while maintaining the quality of importance estimates. We will include a discussion of these future work directions in the revised manuscript.
>
> **[Table 1. Impact of score re-evaluation for ResNet-20 / CIFAR-10 & ResNet-18 / CIFAR-100 with 80% pruning]**
> |**Dataset**|**Re-eval Freq.**|1bit|2bit|4bit|8bit|32bit|**Avg.**|**Re-eval GPU hrs**|**Total GPU hrs**|
> |:-:|:-:|-|-|-|-|-|-|:-:|:-:|
> |CIFAR-10|**- (Ours)**|92.60|93.01|93.03|93.00|93.08|**92.97**|**-**|**1.52**|
> | |**100**|92.08|92.64|92.80|92.82|92.88|**92.75**|**0.38**|**5.31**|
> | |**50**|92.45|92.83|92.96|93.04|92.96|**92.97**|**1.13**|**2.65**|
> | |**30**|92.64|92.90|93.03|92.88|92.69|**92.84**|**2.27**|**3.79**|
> | |**10**|92.21|92.56|92.73|92.86|92.75|**92.73**|**7.33**|**8.85**|
> |CIFAR-100|**- (Ours)**|69.14|70.12|70.35|70.43|70.41|**70.26**|**-**|**1.47**|
> | |**100**|69.16|70.24|70.41|70.39|70.04|**70.18**|**0.77**|**2.24**|
> | |**50**|69.13|70.23|70.60|70.82|70.41|**70.45**|**2.30**|**3.78**|
> | |**30**|68.91|70.17|70.18|70.05|70.09|**70.02**|**4.57**|**6.04**|
> | |**10**|68.95|70.45|70.76|70.78|70.87|**70.66**|**14.57**|**16.04**|
>
> **[Table 2. Impact of score re-evaluation for ResNet-20 / CIFAR-10 & ResNet-18 / CIFAR-100 with 90% pruning]**
> |**Dataset**|**Re-eval Freq.**|1bit|2bit|4bit|8bit|32bit|**Avg.**|**Re-eval GPU hrs**|**Total GPU hrs**|
> |-|-|-|-|-|-|-|-|:-:|:-:|
> |CIFAR-10|**- (Ours)**|92.04|92.66|92.66|92.63|92.38|**92.38**|-|**0.84**|
> ||100|91.43|92.09|92.09|92.09|91.89|**92.09**|**0.37**|**1.21**|
> ||50|91.37|91.98|92.22|92.29|92.32|**92.09**|**1.15**|**1.99**|
> ||30|91.94|92.13|92.12|92.11|92.38|**92.10**|**2.31**|**3.15**|
> ||10|91.24|91.64|91.94|92.03|91.77|**91.82**|**7.33**|**8.17**|
> |CIFAR-100|**- (Ours)**|67.53|69.40|69.75|69.25|69.24|**69.32**|-|**0.83**|
> ||100|67.34|69.17|69.23|69.29|69.28|**69.02**|**0.76**|**1.59**|
> ||50|67.77|69.12|69.43|69.20|69.11|**69.11**|**2.32**|**3.15**|
> ||30|67.82|69.30|69.11|69.34|69.02|**69.15**|**4.63**|**5.46**|
> ||10|67.28|69.14|69.41|69.50|69.44|**69.23**|**14.68**|**15.51**|
>
>
> **W2: Effectiveness of BN sharing for 1-bit quantization**
>
> This is an important point, and we appreciate this insightful observation. In multi-bit settings, weight distributions tend to approximate a normal distribution, where our Bias Correction mechanism works effectively. However, in the 1-bit case, the weight distribution collapses into a near-uniform or binary form, causing a significant distribution shift that Bias Correction alone struggles to adequately address. For that reason, prior works on multi-bit networks often omit the 1-bit setting from their analysis [1,2,3]. In contrast, our method explicitly incorporates this case and demonstrates that assigning a separate BN layer for 1-bit quantization is both a simple and effective solution.
>
> To further support this, we conduct ablation studies comparing two configurations: one where BN is shared across all bit-widths (including 1-bit), and another where the 1-bit case has its own BN, and the remaining bit-widths share a single BN. The results clearly show that including 1-bit in BN sharing significantly degrades 1-bit quantization performance. On the other hand, using a separate BN for 1-bit achieves strong performance with negligible additional overhead, that is the additional parameters for the separate BN layer, which accounts for less than 0.5% of the total number of parameters. This performance degradation stems from unstable BN statistics. The 1-bit weights produce activation distributions that are markedly different from those of higher bit-widths, leading to significant fluctuations in shared BN statistics. These fluctuations negatively impact the normalization of other bit-widths, ultimately harming overall model performance.
>
> **[Table 3. BN sharing with and without 1-bit quantization on ResNet-20 / CIFAR-10]**
> |**Framework**|1bit|2bit|3bit|4bit|5bit|6bit|7bit|8bit|32bit|**Avg.**|
> |-|-|-|-|-|-|-|-|-|-|-|
> |Share BN w/ 1-bit|46.38|91.90|92.57|91.48|90.98|90.53|90.24|90.14|89.90|**86.01**|
> |Share BN w/o 1-bit|92.49|89.44|92.43|92.67|92.43|92.15|92.00|91.97|91.65|**91.91**|
>
>
> **W3+Q3(a): Clarification on the training pipeline**
>
> Thank you for drawing our attention to this, as we fully agree that a more detailed explanation of the training pipeline is in need. For ImageNet-1K experiments, fine-tuning begins from the pretrained Any-Precision model before calibration. Specifically for our method, we use this non-calibrated model as the starting point and apply coreset sampling and Bias Correction during fine-tuning, followed by BN adaptation at the end. In contrast to the original Any-Precision [4], which requires an additional calibration stage after training to achieve comparable accuracy at calibrated bit-widths, our approach eliminates this step. Training concludes after fine-tuning and a lightweight BN adaptation phase.
>
>
> **W3+Q3(b): Provide GPU hours with more detailed granularity**
>
> We truly appreciate the comment and fully agree that providing a more detailed breakdown improves transparency and clarity in evaluating computational efficiency. In response, we report GPU hours for all ResNet experiments, broken down by training stage, including:
>
> - Score evaluation
> - Coreset training
> - Calibration (if applicable)
> - Adaptation
>
> These results also help clarify the computational cost structure of our framework. Specifically, coreset sampling significantly reduces training GPU hours for multi-bit quantization models, and this efficiency gain increases with the pruning rate. However, coreset sampling alone does not eliminate the cost of the calibration phase, which is typically needed to support additional bit-widths. To address this, we apply Bias Correction and BN adaptation, which allow us to remove the calibration step entirely without sacrificing accuracy.
>
> **[Table 4. Breakdown of GPU hours for ResNet-20 experiments with 80% pruning]**
> |Framework|Training|Calibration|Adaptation|Scoring|Total GPU hours (Speedup)|
> |-|:-:|:-:|:-:|:-:|:-:|
> |Dedicated|11.97|-|-|-|11.97 (1.00×)|
> |Any-Prec.|7.51|1.25|-|-|8.76 (1.36×)|
> |Bias Correction (Ours)|7.51|-|0.004|-|7.52 (1.59×)|
> |Bias Correction+Coreset Sampling (Ours)|1.52|-|0.004|0.37 *(Offline)*|1.52 (7.88×)|
>
> **[Table 5. Breakdown of GPU hours for ResNet-18 experiments with 80% pruning]**
> |Framework|Training|Calibration|Adaptation|Scoring|Total GPU hours (Speedup)|
> |-|:-:|:-:|:-:|:-:|:-:|
> |Dedicated|11.19|-|-|-|11.19 (1.00×)|
> |Any-Prec.|7.17|1.10|-|-|8.27 (1.35×)|
> |Bias Correction (Ours)|7.17|-|0.004|-|7.17 (1.56×)|
> |Bias Correction+Coreset Sampling (Ours)|1.47|-|0.004|0.74 *(Offline)*|1.47 (7.61×)|
>
> *Reference*
> [1] K.Xu et al., "Eq-net" ICCV 2023.
> [2] X.Sun et al., "CoQuant" ICCV 2024.
> [3] Y.Zhong et al., "MultiQuant" CoRR 2023.
> [4] H.Yu et al., "Any-precision DNN" AAAI 2021.

---

> ### Comment · Reviewer_Txhn · 2025-08-04
>
> Thank you for addressing my concerns. Given that the key concerns regarding design choice and efficiency have been addressed, I have increased my score and support acceptance

---

### Official Review · Reviewer_iWva · 2025-07-02

**Clarity:** 4
**Significance:** 3
**Originality:** 3
**Rating:** 5
**Confidence:** 4

**Summary:**

This paper focuses on reducing the large training overhead of multi-bit quantization networks, caused by training separate batch-normalization parameters for each bit-width to correct activation distributions and full dataset updates for each bit-width. To that end, the authors propose: 1) weight bias correction, by correcting quantization-induced bias and aligning activation distributions, enabling shared BN across bit-widths, and 2) a bit-wise coreset sampling strategy that accounts for sample importance, obtained via a novel bit-wise training scheme, across bit-width and training epoch. The method is demonstrated using multiple datasets and models and achieves consistent GPU hours speedup while matching the performance of other works.

**Questions:**

1. How important is the BN adaptation performed at the last training stage? Why is bias correction for quantized weights not enough for activation alignment? Are most of the accuracy gains due to BN adaptation or bias correction?
2. How often is coreset sampling performed? What is the impact of the frequency at which it is performed on accuracy and GPU hours? Also, from the results, it seems that most of the speedup is due to the coreset sampling method. What would the accuracy and speedup be if only coreset sampling is performed?
3. In Section 3, how is the statement “achieving optimal performance without additional overhead” true? It is clear that BN adaptation will incur additional overhead that needs to be considered.
4. For Table 3 and 4, it is mentioned that the results are obtained by fine tuning a pretrained Any-Precision model. Why are the GPU hours for Any-Precision model training then not included in the reported hours for the proposed method?
5. Some results in Table 1, 3 and 5 are bolded while they are not the best. Why?

**Ethical Concerns:**

["NO or VERY MINOR ethics concerns only"]

**Limitations:**

Yes.

**Quality:**

3

**Strengths And Weaknesses:**

Strengths:
1. The paper is well written and provides clear background information.
2. The observations made throughout the paper are deeply insightful and clearly motivate the solution approach.
3. The paper introduces a novel bit-wise training scheme for score evaluation that addresses the limitations of traditional methods aimed at single precision networks.
4. The results are positive, showcasing a reduction in GPU hours while matching the performance of existing methods.

Weaknesses:
1. Activation alignment is showcased for the inefficient separate BN parameters method. However, the authors do not show the alignment achieved by the proposed bias correction for quantized weights method making the impact of the method unclear.
2. The impact of BN adaptation at the final training stage is not clear. Ablation studies are needed to determine how much the alignment and final accuracy is impacted by bias correction for quantized weights vs. BN adaptation.
3. Resampling is not addressed in the results. While it has been shown that sample importance changes over training epochs, the paper does not provide guidelines on how to choose the frequency of coreset sampling and its impact on accuracy and GPU hours.
4. Some of the figures use symbols that are not explained anywhere, making it difficult to understand (ex. Figure 1 with D and S)

---

> ### Author Rebuttal · Authors · 2025-07-31
>
> We deeply thank the reviewer for their constructive and detailed feedback, and also for recognizing the insights and empirical results of our work. Below, we provide point-by-point responses to the concerns and questions raised.
>
> **W1: Alignment achieved by Bias Correction**
>
> Thank you for pointing this out. To demonstrate the effectiveness of our Bias Correction method, we conducted an analysis of the output activations, comparing how well activations from different bit-widths align with one another. In all experiments, the activation precision is fixed to 4 bits, and we measure the mean absolute error (MAE) between the activations of quantized and full-precision models, with and without applying Bias Correction. The results show that applying the correction consistently improves alignment, reducing the avg. MAE from 0.755 to 0.667. This improvement holds across all bit-widths, demonstrating our method’s effectiveness. We will include these results in the revised version to highlight the impact of Bias Correction on activation alignment.
>
> **[Table 1. MAE between different bit-widths and full precision models' activation with vs. without Bias Correction]**
> |Bit-width|With Bias Correction|Without Bias Correction|
> |-|:-:|:-:|
> |2bit|0.667|0.744|
> |3bit|0.670|0.756|
> |4bit|0.671|0.754|
> |5bit|0.654|0.750|
> |6bit|0.683|0.761|
> |7bit|0.668|0.756|
> |8bit|0.659|0.766|
> |**Avg.**|**0.667**|**0.755**|
>
>
> **W2+Q1: Clarify the role and relative impact of Bias Correction vs. BN adaptation**
>
> Thank you for this point. As shown in Table 2, our ablation studies isolate each component to assess its individual contribution to performance. The results reveal that neither method alone is sufficient for optimal accuracy across all bit-widths. For example, using only Bias Correction leads to suboptimal performance at 2-bit, while using only BN adaptation underperforms at 1-bit. Using both techniques together consistently yields the best accuracy across all bit-widths.
>
> While Bias Correction offers a low-cost way to compensate for systematic errors in the weight space, it does not directly influence the BN running statistics that govern activation distributions at inference time. BN adaptation, on the other hand, recalibrates these statistics through forward passes on the training data, directly aligning the activation distributions. In this sense, BN adaptation complements Bias Correction by targeting residual discrepancies that remain after correcting the weights. Together, the two methods correct both weight distribution shifts and activation misalignments, enabling consistent activation behavior and improved accuracy, particularly for intermediate calibrated bit-widths (e.g., 3,5,6,7 bits). We plan to include these ablation studies in the revised version to more clearly demonstrate the distinct and complementary roles of each technique.
>
> **[Table 2. The impact of Bias Correction vs. BN adaptation]**
> |Dataset|Bias Correction|BN adaptation|1bit|2bit|3bit|4bit|5bit|6bit|7bit|8bit|32bit|Avg.|
> |-|:-:|:-:|-|-|-|-|-|-|-|-|-|-|
> |CIFAR-10|||92.95|87.72|93.53|93.32|92.73|92.47|92.08|91.87|93.53|92.24|
> ||✔||93.58|91.98|93.50|93.68|93.62|93.51|93.47|93.40|93.70|93.38|
> |||✔|92.87|93.36|93.59|93.55|93.56|93.52|93.56|93.63|93.65|93.48|
> ||✔|✔|93.61|93.72|93.88|93.89|93.92|93.84|93.88|93.83|93.92|93.83|
> |CIFAR-100|||70.23|53.18|70.48|70.88|69.52|68.19|67.25|66.88|70.21|67.42|
> ||✔||71.12|69.03|71.58|72.03|71.63|71.39|71.23|71.10|71.83|71.22|
> |||✔|70.36|70.95|71.53|71.56|71.40|71.46|71.52|71.47|71.45|71.30|
> ||✔|✔|71.37|72.10|72.31|72.37|72.27|72.34|72.38|72.33|72.26|72.19|
>
> **W3+Q2: Impact of resampling frequency**
>
> We agree that sampling frequency is an important factor to examine. In practice, the overhead of bit-wise coreset resampling is extremely small compared to the overall training cost.  For example, even on an ImageNet-scale dataset, performing 100 resamplings takes only about 3.36 minutes. Given this negligible cost, resampling at every epoch is a practical and effective choice.
>
> To quantitatively demonstrate this, we conducted experiments with different resampling intervals and measured both validation accuracy and total sampling time. The results show that resampling every epoch improves average accuracy by 1.33%p compared to resampling every 30 epochs, while adding just 53 seconds of overhead to a multi-hour training process. This demonstrates that frequent resampling can offer meaningful accuracy gains at virtually no additional cost.
>
> **[Table 3. Influence of sampling frequency on ResNet-20 / CIFAR-10]**
> |**Resampling Frequency**|1bit|2bit|4bit|8bit|32bit|**Avg.**|**Total Sampling Time (% of Total GPU time)**|
> |-|-|-|-|-|-|-|:-:|
> |1 **(Ours)**|92.60|93.01|93.03|93.00|93.08|**92.97**|53.02s **(0.96%)**|
> |10|91.96|92.40|92.65|92.66|92.75|**92.55**|5.09s **(0.09%)**|
> |20|91.40|91.73|92.04|92.03|92.34|**91.98**|2.77s **(0.05%)**|
> |30|91.29|91.53|91.63|91.73|91.68|**91.64**|1.40s **(0.03%)**|
>
> **Q2: Impact of coreset sampling**
>
> Thank you for raising this point. To examine the standalone contribution of coreset sampling to both accuracy and training speedup, we provide results on a CIFAR-10 baseline where only coreset sampling is applied, with all bit-widths except for 1-bit sharing BN layers.
>
> In the table below, we denote setups with coreset sampling as A, and those with Bias Correction and BN adaptation as B. The configuration A only (B not applied) represents coreset sampling alone, while A+B combines coreset sampling with Bias Correction and BN adaptation. As shown in the results, while coreset sampling (A) contributes most to the speedup, it is not sufficient on its own to maintain strong accuracy at across every bit-width. In contrast, when Bias Correction and BN adaptation are applied alongside coreset sampling (A+B), we observe accuracy improvements everywhere, with pratically no additional GPU hours. The results indicate that although coreset sampling is the main driver of compute efficiency, Bias Correction and BN adaptation is essential for best accuracy.
>
> **[Table 4. Impact of only coreset sampling on ResNet-20 / CIFAR-10 (A: Coreset Sampling, B: Bias Correction & BN adaption); Bold the highest performance for each pruning rate.]**
> |**A (Pruning Ratio)**|**B**|1bit|2bit|3bit|4bit|5bit|6bit|7bit|8bit|32bit|Avg.|**GPU hrs (Speed Up)**|
> |-|-|-|-|-|-|-|-|-|-|-|-|:-:|
> |X (0.0)|X|92.95|87.72|93.53|93.32|92.73|92.47|92.08|91.87|93.53|92.24|7.51 (1.00×)|
> |O (0.7)|X|92.46|87.42|92.92|93.05|92.51|91.95|91.69|91.49|92.98|91.83|2.06 (3.65×)|
> |O|O|**92.62**|**93.03**|**92.98**|**93.08**|**93.05**|**93.05**|**93.08**|**93.10**|**93.15**|**93.02**|2.06 (3.65×)|
> |O (0.8)|X|92.29|87.61|92.84|92.27|91.51|90.99|90.68|90.46|92.84|91.28|1.52 (4.94×)|
> |O|O|**92.60**|**93.01**|**92.96**|**93.03**|**93.02**|**92.99**|**93.01**|**93.00**|**93.08**|**92.97**|1.52 (4.94×)|
> |O (0.9)|X|91.47|87.07|91.61|91.22|90.42|89.62|89.23|89.00|91.62|90.14|0.83 (9.05×)|
> |O|O|**92.04**|**92.66**|**92.61**|**92.66**|**92.47**|**92.50**|**92.51**|**92.63**|**92.38**|**92.50**|0.83 (9.05×)|
>
> **W4: Miscellaneous figure errors**
>
> Thank you for drawing our attention to this. For Figure 1, we will clarify that D and S denote the full dataset and the sampled coreset, respectively. For Figure 2, we will update the caption to explicitly define E[X] and V[X] as the running statistics of BN layers before adaptation, and E[X]’ and V[X]’ as those after the BN adaptation phase. These changes will be reflected in the revised version to improve clarity of our presentation.
>
> **Q3: Overhead of BN Adaptation**
>
> While BN adaptation does introduce additional computation, its overhead is negligible compared to the overall training cost since it involves only forward passes over a fixed number of samples and does not require backpropagation. For example, in the ResNet-20 with CIFAR-10 setting, BN adaptation takes only 0.004 GPU hours, accounting for less than 0.3% of the total training time in both the Bias Correction-only configuration (7.52 GPU hours) and the more efficient Bias Correction + Coreset Sampling configuration (1.52 GPU hours). Below, we include a detailed breakdown of GPU hours across all components, highlighting the negligeable overhead of BN adaptation step.
>
> **[Table 5. Breakdown of GPU hours for ResNet-20 / CIFAR-10 experiments with 80% pruning]**
> |Framework|Training|Calibration|Adaptation|Scoring|Total GPU hours (Speedup)|
> |-|:-:|:-:|:-:|:-:|:-:|
> |Dedicated|11.97|-|-|-|11.97 (1.00×)|
> |Any-Prec.|7.51|1.25|-|-|8.76 (1.36×)|
> |Bias Correction|7.51|-|**0.004**|-|7.52 (1.59×)|
> |Bias Correction+Coreset Sampling|1.52|-|**0.004**|0.37 *(Offline)*|1.52 **(7.88×)**|
>
>
> **Q4: Pretraining settings for ImageNet-1K experiments**
>
> Thank you for raising this important question. In our experiments, we fine-tune from publicly available Any-Precision ResNet-50 checkpoints [1], rather than training from scratch. This follows prior work, as full Any-Precision training on ImageNet-1K is extremely time-consuming, typically requiring around 300 GPU hours for a single run—making it infeasible to reproduce during the rebuttal period. To our knowledge, no existing multi-bit quantization work has explicitly reported this pretraining cost. Accordingly, the GPU hours reported in Tables 3 and 4 include only the additional fine-tuning and calibration time. However, we agree that including full pretraining costs would improve clarity and will report those figures in the revised manuscript.
>
> **Q5: Bolded results in tables**
>
> The bolded values are meant to indicate results from our proposed method. We apologize that this was not clearly stated and will revise them accordingly. Additionally, we will consider using a different visual cue to distinguish our method from others more clearly.
>
> *Reference*
> [1] H.Yu et al., "Any-precision DNN." AAAI 2021.

---

> > ### Comment · Reviewer_iWva · 2025-08-05
> >
> > I have read the authors' rebuttal to my comments and find them satisfactory. My rating remains unaltered.

---

### Comment · Area_Chair_97aS · 2025-08-03

Dear reviewers,

We are halfway through the author-reviewer discussion period. If you haven't already, please review the rebuttal (and other reviews) to check if the authors have addressed your concerns, acknowledge to the authors that you have read their response, and make any necessary adjustments to the score as needed. Thanks

---

### Note · Authors · 2025-08-15

We thank all the ACs and reviewers for their engagement, which has greatly improved the clarity and relevance of our work. Below we summarize three key themes that recurred in the reviews.

**1. Identifying contributions of each component in our method**

To identify the distinct roles of Bias Correction, BN adaptation, and coreset sampling, we conducted detailed ablation studies that quantified each component’s contribution to overall accuracy and efficiency. The results show that coreset sampling primarily drives training speedup, while Bias Correction and BN adaptation together ensure stable accuracy across all bit-widths.

**2. Strengthening theoretical and empirical foundations**

We provided the theoretical basis of Bias Correction, showing that aligning quantized weight statistics can effectively reduce activation mismatches. We also detailed implicit cross-bit-width knowledge transfer, where shared weights across quantization levels act as a form of indirect supervision. Additional experiments on BN sharing, sampling frequency, and dynamic score re-evaluation further confirm our design choice, consistently matching or surpassing alternative strategies with greater efficiency.

**3. Demonstrating efficiency gains and practical integration**

We added detailed GPU-hour breakdowns for all configurations, compared our cost model to the baselines, and quantified the negligible overhead of BN adaptation and resampling. We also clarified the training pipeline and will release code and integration guides to enable straightforward adoption in existing workflows.

Our work provides a practical and well-founded framework for training multi-bit networks. With Bias Correction and bit-wise coreset sampling, our method removes the need for costly post-training calibration and significantly reduces training overhead, enabling flexible models that adapt to diverse hardware constraints and can be readily deployed in resource-limited environments.

Through additional experiments and clarifications, we addressed all reviewer comments. Feedback has been consistently positive, with one reviewer raising their score and all reviewers now recommending acceptance. Accordingly, we see no lingering concerns that should preclude acceptance, and given this consensus, we believe the strengthened manuscript is well positioned to make meaningful contributions to the community. Once again, we thank all the ACs and reviewers for their contributions to improving this manuscript.

---

### Decision · Program_Chairs · 2025-09-17

**Decision:**

Accept (poster)

**Comment:**

This paper introduces techniques to reduce the training overhead of multi-bit quantization networks, including weight bias correct enabling shared batch normalization and bit-wise coreset sampling which allows each child model to train on a compact subset. The methods are validated on CNN and Vision Transformer models, showing clear speedup while matching the accuracy of prior methods.

The paper is very well written, with observation, problem statements, and solutions clearly motivated, explained, and insightful. The proposed methods are reasonably novel. The results demonstrate significant improvement in training time over existing methods while preserving model quality.

During the rebuttal, the authors provided additional data and clarification, such as the impact of resampling frequency, BN adaptation, the computation overhead for dynamic sampling. All reviewers gave positive evaluation. Therefore, this paper is recommended for acceptance.